# FedVLMBench: Benchmarking Federated Fine-Tuning of Vision-Language Models

## Abstract

Vision-Language Models (VLMs) have demonstrated remarkable capabilities in cross-modal understanding and generation by integrating visual and textual information. While instruction tuning and parameter-efficient fine-tuning methods have substantially improved the generalization of VLMs, most existing approaches rely on centralized training, posing challenges for deployment in domains with strict privacy requirements like healthcare. Recent efforts have introduced Federated Learning (FL) into VLM fine-tuning to address these privacy concerns, yet comprehensive benchmarks for evaluating federated fine-tuning strategies, model architectures, and task generalization remain lacking. In this work, we present **FedVLMBench**, the first systematic benchmark for federated fine-tuning of VLMs. FedVLMBench integrates two mainstream VLM architectures (encoder-based and encoder-free), four fine-tuning strategies, seven FL algorithms, six multimodal datasets spanning four cross-domain single-task scenarios and two cross-domain multitask settings, covering four distinct downstream task categories. Through extensive experiments, we uncover key insights into the interplay between VLM architectures, fine-tuning strategies, data heterogeneity, and multi-task federated optimization. Notably, we find that a 2-layer multilayer perceptron (MLP) connector, along with concurrent connector-LLM tuning emerges as the optimal configuration for encoder-based VLMs in FL. Furthermore, current FL methods exhibit significantly higher sensitivity to data heterogeneity in vision-centric tasks than text-centric ones, across both encoder-free and encoder-based VLM architectures. Our benchmark provides essential tools, datasets, and empirical guidance for the research community, offering a standardized platform to advance privacy-preserving, federated training of multimodal foundation models. Our dataset and code could be found at this anonymous link.

## 1 Introduction

Vision-Language Models (VLMs) (Achiam et al., 2023; Liu et al., 2023; Team et al., 2023) have demonstrated groundbreaking advancements in cross-modal understanding and generation tasks by integrating multimodal vision and language information. Instruction tuning methods (e.g., LLaMA-Adapter V2 (Gao et al., 2023)), and parameter-efficient tuning techniques (e.g., LoRA (Hu et al., 2022)) further improve these models' generalization to unseen tasks and scenarios. This characteristic positions VLMs as a potential foundational architecture for addressing complex open-domain tasks, including privacy-preserving scenarios. For example, in the medical domain, several works (Moor et al., 2023; Li et al., 2023; Zhu et al., 2023; Pan et al., 2025) have successfully integrated visual and language information using few-shot learning or reinforcement learning to enhance the analytical capabilities of VLMs in tasks such as report generation. However, most VLM instruction-tuning methods (Hu et al., 2022; Liu et al., 2023) typically adopt a centralized learning paradigm, which fails to meet the privacy protection requirements necessary for sensitive fields like healthcare and finance. While recent research (Xu et al., 2025; Zhang et al., 2025; Miao et al., 2025; Ghiasvand et al., 2025) has introduced FL into the instruction fine-tuning of VLMs to effectively address data privacy concerns, significant limitations remain.

First, existing VLMs can be categorized into two main types, encoder-based VLMs (Team, 2025; Liu et al., 2023) and encoder-free VLMs (Li et al., 2024; Xie et al., 2024), depending on the inclusion of visual encoders. These paradigms differ fundamentally in their components and fine-tuning mechanics.

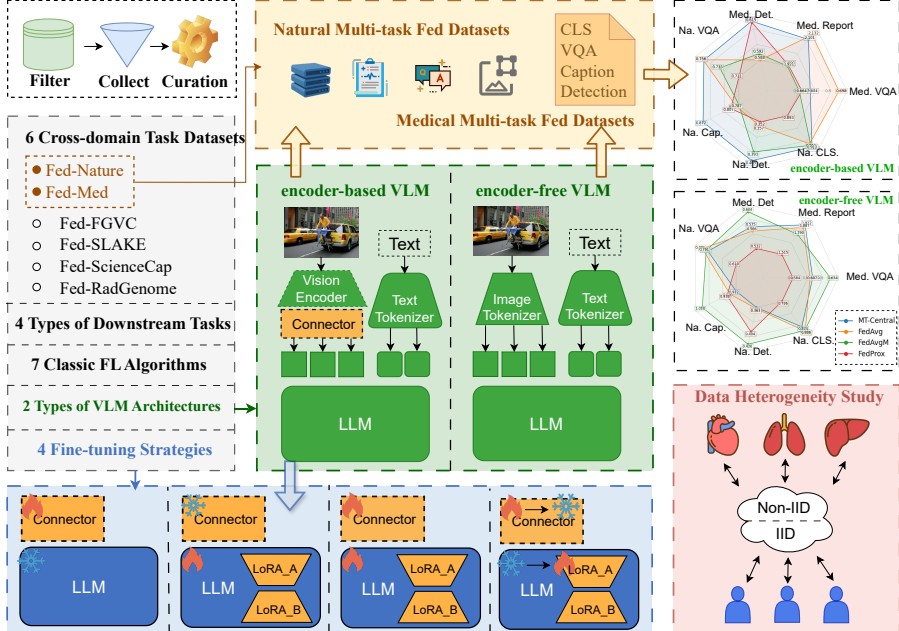

Figure 1: Overview of FedVLMBench, which integrates 2 types of mainstream VLM architectures, 4 fine-tuning strategies, 7 FL algorithms, and 6 cross-domain task datasets. This framework facilitates comprehensive evaluation and comparison of multitask learning approaches in FL contexts.

Encoder-based models typically involve a vision encoder, a connector, and an LLM, leading to diverse and complex fine-tuning strategies (e.g., tuning only the connector, concurrently tuning the connector and LLM, etc). In contrast, encoder-free architectures, by omitting the separate encoder, often rely on alternative parameter-efficient tuning methods. Current research (Xiong et al., 2025; Saha et al., 2025; Zhang et al., 2025) in FL primarily focuses on designing client-specific adapters or new aggregation methods based on encoder-based VLM, but but largely overlooks the emerging encoder-free paradigm and the extensive spectrum of fine-tuning strategies relevant to each architecture.. This narrow perspective fails to address how differences in VLM architecture and fine-tuning strategies interact with data heterogeneity in FL, leaving a critical gap in understanding how to effectively apply FL across varied VLMs. Second, existing FL multimodal benchmark research focuses narrowly on two basic task types—Visual Question Answering (VQA) and classification—while ignoring more complex but critically important multimodal tasks such as report generation and visual localization (Tab. 1). Third, no existing FL datasets support federated multi-modal multi-task learning scenarios, despite their practical significance in real-world applications where different clients may need to handle distinct multimodal tasks (e.g., one hospital specializes in classification while another focuses on report generation). To address these research gaps, this paper shifts from technical improvements in existing federated instruction tuning methods to exploring three core foundational questions:

**Q1**: How do choices in connector layer design and fine-tuning strategies impact the performance of federated finetuning of encoder-based VLMs across diverse single-task FL scenarios?

**Q2**: How do different FL algorithms, using both encoder-based and encoder-free VLMs as baseline architectures, perform under varying data heterogeneity conditions in single-task federated fine-tuning processes?

**Q3**: How do different FL algorithms, using both encoder-based and encoder-free VLMs as baseline architectures, perform in practical multi-modal multi-task FL environments?

To systematically address these questions, we developed an innovative FL fine-tuning of VLMs benchmark **FedVLMBench** that integrates **2** types of mainstream VLM architectures (encoder-based and encoder-free VLMs), **4** fine-tuning strategies, **7** FL algorithms, **4** types of downstream tasks, and **6** cross-domain task datasets. As shown in Tab. 1, our benchmark differs from existing works by encompassing a broader range of downstream tasks, diverse VLM architectures, and unique multi-task collaborative fine-tuning datasets. Through extensive experimental analysis, we present the following key findings:

Table 1: Comparisons of FedVLMBench with other FL benchmarks. # Collab. Datasets refer to the number of multi-task collaborative fine-tuning datasets.

| Benchmark | Language | Vision | # Arch. Types | # Task Types | # Datasets | # Collab. Datasets |
|---|---|---|---|---|---|---|
| FS-LLM (Kuang et al., 2024) | ✓ | ✗ | 1 | 2 | 3 | ✗ |
| FedLLM-Bench (Ye et al., 2024a) | ✓ | ✗ | 1 | 2 | 4 | ✗ |
| OpenFedLLM (Ye et al., 2024b) | ✓ | ✗ | 1 | 2 | 8 | ✗ |
| FedMLLM (Xu et al., 2025) | ✓ | ✓ | 1 | 2 | 5 | ✗ |
| **FedVLMBench** (Ours) | ✓ | ✓ | **2** | **4** | 6 | **2** |

1) For encoder-based VLM in FL, a 2-layer MLP connector stands out as the most effective connector when compared to other linear or more complex MLP configurations; concurrent fine-tuning both the connector and the LLM yields superior task-agnostic performance compared to the sequential approach of fine-tuning the connector first and then the LLM, while maintaining computational efficiency.

2) For encoder-based VLMs in FL, text-centric tasks (such as VQA and caption generation) benefit dominantly from LLM fine-tuning, while connector fine-tuning should be prioritized for vision-centric tasks like classification and detection.

3) Current FL optimization methods are ineffective for both encoder-free and encoder-based VLMs when dealing with non-IID data partitions in single vision-centric FL tasks, calling for novel solutions addressing the data heterogeneity challenges present in vision-centric FL tasks.

4) While single-task FL struggles with vision-centric performance degradation under non-IID data, federated multitask training achieves near-ceiling performance comparable to centralized training across both text- and vision-centric tasks, regardless of VLM architectures.

The main contributions of this paper can be summarized as follows:

1. We propose **FedVLMBench**, the first systematic benchmark for federated fine-tuning of VLMs. It integrates two mainstream VLM architectures (encoder-based and encoder-free), four fine-tuning strategies, seven diverse FL algorithms, and six cross-domain datasets spanning task categories from text-centric (VQA/captioning) to vision-centric (classification/detection), while comprehensively supporting both single-task and multi-task FL scenarios.

2. We bridge critical gaps in FL benchmarks by introducing (i) four cross-domain single-task datasets with configurable IID, simulated non-IID, and real-world non-IID data distributions, and (ii) two novel multi-task vision-language datasets reflecting real-world non-IID scenarios where clients handle distinct yet interconnected tasks.

3. Through comprehensive evaluation on FedVLMBench, we establish actionable guidelines for federated fine-tuning of VLMs and reveal open challenges for future research in FL multimodal systems.

## 2 RELATED WORK

**Vision-Language Models** (VLMs) (Achiam et al., 2023; Team et al., 2023; Lin et al., 2024) have rapidly advanced by significantly enhancing perceptual and reasoning capabilities through the integration of multimodal information, including text, images, and video. Currently, VLMs can be categorized into two primary types: encoder-based models and encoder-free models. The former encompasses models such as LLAVA (Liu et al., 2023), which utilize pretrained encoders (e.g., CLIP (Radford et al., 2021)) to extract multimodal features and integrate them with LLMs for executing complex tasks. In contrast, encoder-free models (Li et al., 2024; Xie et al., 2024) directly tokenize multimodal data, such as images, enabling adaptive processing of diverse inputs and enhancing the generalizability of VLMs.

**Federated Learning** (FL) (McMahan et al., 2017; Zeng et al., 2024b; Zhang et al., 2024c; Guo et al., 2025a;b) is a privacy-preserving distributed training paradigm that facilitates collaborative modeling through client-localized data processing. The traditional FedAvg (McMahan et al., 2017) method relies on client data volume for parameter-weighted fusion but often suffers from performance degradation in non-IID scenarios. To address this, various optimization schemes have been proposed, such as FedProx (Li et al., 2020), FedAdagrad, FedAdam, and FedYogi (Reddi et al., 2020), PerAvg (Fallah et al., 2020), and FedTGP (Zhang et al., 2024b). More recently, researchers have begun exploring FL in the context of multimodal learning, such as FedLPS (Jia et al., 2024), FedMBridge (Chen & Zhang, 2024), and Pilot (Xiong et al., 2025). For example, Pilot tackles the reduction in VLM generalization by using dynamic adapter designs and a globally shared semantic space.

Table 2: Statistics of 6 federated multimodal fine-tuning datasets in FedVLMBench.

| Dataset | Task Type | Data Source | Data Type | #Max Real Clients | #Instances | Evaluate metric |
|---|---|---|---|---|---|---|
| Fed-FGVC | CLS | FGVC (Maji et al., 2013) | Image | 30 | 9,967 | Acc |
| Fed-ScienceCap | Caption Generation | ScienceQA (Lu et al., 2022) | Image+Text | 27 | 5,157 | CIDER/ROUGE_L |
| Fed-SLAKE | VQA | SLAKE (Liu et al., 2021) | Image+Text | 3 | 8,061 | Acc |
| Fed-RadGenome | Detection | RadGenome-Chest CT (Zhang et al., 2024d) | Image+Text | 3 | 8,744 | IoU |
| Fed-Nature | VQA | COCO-QA (Ren et al., 2015) | Image+Text |  | 6,000 | Acc |
|  | Visual Grounding | RefCOCO (Kazemzadeh et al., 2014) | Image+Text | 4 | 6,000 | IoU |
|  | Caption Generation | RefCOCO | Image+Text |  | 6,000 | CIDER/ROUGE_L |
|  | CLS | COCO (Lin et al., 2015) | Image |  | 6,000 | Acc |
| Fed-Med | VQA | SLAKE & VQA-RAD (Lau et al., 2018) | Image+Text |  | 3,846 | Acc |
|  | Detection | RadGenome-Chest CT | Image+Text | 3 | 8,744 | IoU |
|  | Report Generation | MIMIC-CXR (Johnson et al., 2019) | Image+Text |  | 8,000 | CIDER/ROUGE_L |

FedMLLM (Xu et al., 2025) introduces a benchmark for evaluating federated fine-tuning performance of MLLMs across heterogeneous scenarios. However, these approaches do not systematically explore critical issues such as vision language model architecture, the interplay of different modules, and the intricacies of multi-task collaborative training within the FL context.

## 3 FEDERATED VISION-LANGUAGE BENCHMARK DATASETS

Current federated benchmarks (Xu et al., 2025) exhibit two fundamental limitations in task coverage. First, existing works predominantly focus on only two basic task types—VQA and classification, while overlooking more complex yet critically important multimodal tasks, such as report generation and detection. Second, and more importantly, there exists a complete absence of datasets supporting federated multi-modal multi-task learning scenarios, despite their practical significance in real-world applications where different clients may need to handle distinct multimodal tasks. To bridge the gaps, we develop six novel federated datasets through two synergistic efforts. On the single-task front, we construct four specialized benchmarks (Fed-FGVC, Fed-SLAKE, Fed-ScienceCap, and Fed-RadGenome) that significantly expand beyond conventional VQA and classification to include caption generation and detection tasks, with careful consideration of both IID and non-IID data distributions. More innovatively, we pioneer two multi-task federated datasets (Fed-Nature and Fed-Med) that, for the first time, enable collaborative instruction tuning across interconnected multi-task and multimodal objectives, thereby filling a crucial void in current FL research infrastructure.

**Fed-FGVC: A Classification Vision-Language FL Dataset.** FGVC-Aircraft (Maji et al., 2013) is a dataset designed for fine-grained visual classification of aircraft. Based on the key attribute "manufacturer", we distribute the data among up to 30 clients, ensuring that every three categories are evenly distributed or merged, resulting in IID and non-IID partitions. Additionally, four heterogeneous partitions are generated using varying Dirichlet coefficients, resulting in a Fed-FGVC dataset with six partitions to benchmark multimodal language models on fine-grained image understanding.

**Fed-ScienceCap: A Caption Generation Vision-Language FL Dataset.** ScienceQA (Lu et al., 2022) is a comprehensive dataset encompassing various question types from real science exams across different disciplines. We screened image-description pairs and excluded categories with fewer than 100 samples by "category". The remaining 27 categories were either evenly distributed or merged to create a maximum of 27 clients, facilitating the development of IID and non-IID partitions. The resulting Fed-ScienceCap dataset offers two partitioning schemes to evaluate models on image semantic understanding in the natural sciences.

**Fed-SLAKE: A Visual Question Answering Vision-Language FL Dataset.** SLAKE (Liu et al., 2021) is a dataset for medical vision problems, covering various modalities, organs, and both closed and open questions. We first excluded question types with fewer than 20 samples and then used uniform and complete partitioning by "modality" to create IID and non-IID partitions among 3 clients, resulting in the Fed-SLAKE dataset.

**Fed-RadGenome: A Visual Detection Vision-Language FL Dataset.** RadGenome-Chest CT (Zhang et al., 2024d) is a multimodal dataset containing segmentation masks and region-specific reports for 3D chest CT scans. We extracted two 2D cross-sectional images from each 3D volume, along with masks for three organs (heart, lung, and abdomen) and their corresponding reports. Using uniform and complete category division methods, we distributed the data among 3 clients, resulting in the Fed-RadGenome dataset, which includes over 8,000 samples and both IID and non-IID partitions.

**Fed-Nature: A Natural Multitask Vision-Language FL Dataset.** Fed-Nature integrates three public vision-language datasets — COCO (Lin et al., 2015) (classification), RefCOCO (Kazemzadeh et al., 2014) (visual grounding and captioning generation), and COCO-QA (Ren et al., 2015) (VQA) — by linking their cross-modal annotations through shared image IDs. We map each specific task to a

dedicated client, creating four clients that jointly support VQA, classification, visual grounding, and caption generation tasks.

**Fed-Med: A Medical Multitask Vision-Language FL Dataset.** Fed-Med unifies chest-related medical question answering, detection, report generation, and various other data sourced from the SLAKE (Liu et al., 2021) (VQA), MIMIC-CXR (Johnson et al., 2019) (report generation), VQA-RAD (VQA) (Lau et al., 2018), and RadGenome-Chest CT (Zhang et al., 2024d) (detection) datasets. Similar to Fed-Nature, we map each specific task to a client, creating three clients that jointly support VQA, report generation, and detection.

More details about the datasets and their partitions are provided in the Appendix B and Appendix C.3.

## 4 FEDVLMBENCH FRAMEWORK

To make our FedVLMBench framework compatible with standard FL protocols, it follows the same training process as conventional FL (e.g., FedAvg (McMahan et al., 2017)), which involves a central server and $K$ clients. Each client holds a private multimodal dataset $D_k = \{(I^{(i)}, T^{(i)}, Res^{(i)}) \mid i = 1, 2, \ldots, N_k\}$ that includes images $I$, text $T$, and corresponding responses $Res$. The underlying optimization goal of our FedVLMBench can be formalized as follows:

$$arg \min_{w^s \in \mathbb{R}^d} \frac{1}{K} \sum_{k=1}^{K} \mathcal{L}_{\text{VLM}}^{(k)}(w_k), \tag{1}$$

where $\mathcal{L}_{\text{VLM}}^{(k)}(w_k)$ denotes the local loss function of client $k$, $N_k$ represents the number of samples in client $k$'s private dataset, $w_k$ represents the entire model parameters of client $k$, and $w^s$ denotes the trainable parameters.

Our FedVLMBench framework, as illustrated in Fig. 1, involves two mainstream VLM architectures: encoder-based and encoder-free VLMs. The former utilizes a connector $\mathcal{C}(\cdot; \theta_c)$ to map features extracted from the image encoder $\mathcal{E}$ into tokens, while the encoder-free approach directly employs the image tokenizer $\mathcal{T}_{\text{img}}$ to generate tokens. Both models use the text tokenizer $\mathcal{T}_{\text{text}}$ to encode textual information. For the encoder-based VLM, we employ four fine-tuning strategies that explore different orders and combinations of fine-tuning the connectors and LLMs. Specifically, the first strategy focuses on fine-tuning only the connector. The second strategy involves fine-tuning only the LLM using LoRA (Hu et al., 2022). The third strategy entails simultaneously fine-tuning both the connector and the LLM with LoRA. Finally, the fourth strategy consists of fine-tuning the connector first, followed by the LLM using LoRA. For the encoder-free VLM, we only utilize LoRA to fine-tune the LLM.

In each FL communication round, the server first broadcasts the trainable parameters to each client. Then, clients conduct local fine-tuning and share the updated weights with the server for aggregation. The server aggregates these updates to update the global model and then re-broadcasts the trainable parameters to each client for the next round of fine-tuning. We will elaborate on this workflow in the following.

**Local Fine-Tuning Procedure.** For each round of local fine-tuning, we first update the trainable parameters with the received parameters, which may be partial due to varying training strategies. Then we perform stochastic gradient descent steps to update the trainable parameters. The update process is shown below:

$$w_k^s \leftarrow w_k^s - \eta_g \nabla_{w_k} \mathcal{L}_{\text{VLM}}^{(k)}(w_k), \tag{2}$$

where $w_k^s$ represents the trainable parameters of client $k$. For the encoder-based VLM, its composition varies according to the different fine-tuning strategies:

$$w_k^s = \begin{cases} \theta_c, & \text{fine-tune only the connector,} \\ \theta_{\text{LLM}}, & \text{fine-tune only the LLM using LoRA,} \\ \{\theta_c, \theta_{\text{LLM}}\}, & \text{fine-tune both the connector and LLM with LoRA simultaneously,} \\ \{\theta_c, \theta_{\text{LLM}}\}, & \text{fine-tune the connector and LLM with LoRA in order,} \end{cases} \quad (3)$$

where $\theta_c$ and $\theta_{LLM}$ represent the trainable parameters of the connector and LoRA in LLM, respectively. For the encoder-free VLM, we utilize LoRA to only fine-tune the parameters of the LLM, thus $w_k^s = \theta_{LLM}$.

**Global Aggregation.** Similar to common FL algorithms, the server performs weighted averaging of the trainable parameters as:

$$\bar{w}^s = \sum_{k=1}^{K} \alpha_k w_k^s, \quad (4)$$

where $\alpha_k$ is the aggregation weight for client $k$. In FedAvg, this weight is typically determined by the number of samples at the client, i.e., $\alpha_k = \frac{N_k}{\sum_{k=1}^{K} N_k}$.

## 5 EXPERIMENTS

We systematically investigate federated fine-tuning of VLMs through three progressive dimensions. First, we explore efficient fine-tuning of encoder-based VLMs in FL environments, evaluating the effects of different connector layers (linear, 2-layer MLP, 6-layer MLP) and fine-tuning strategies under IID and non-IID data distributions. Next, we compare encoder-based and encoder-free VLMs to uncover architectural differences in handling data heterogeneity and task-specific sensitivities in single-task FL. Building on these single-task FL findings, we then evaluate federated multitask learning under both encoder-free and encoder-based VLMs.

### 5.1 EXPERIMENTAL SETUP

**Implement Details.**

For encoder-based VLM, we adopt LLaVA 1.5's architecture, utilizing a pre-trained CLIP visual encoder (ViT-B/32 (Dosovitskiy et al., 2021; Radford et al., 2021)) for visual feature extraction and LLAMA3.2-3B (Meta, 2024) as the language model. We investigate three connector layer configurations between visual and language modules: linear layer, 2-layer MLP and 6-layer MLP. For encoder-free VLMs, we initialize pre-trained Show-O-3B (Xie et al., 2024) for instruction fine-tuning. Additionally, we also incorporate Qwen2.5-VL-3B , Qwen2.5-VL-7B (Bai et al., 2025) and and Mono-InternVL-3B (Luo et al., 2025) in our analysis. Across both architectures, we employ LoRA with ranks of 8 and 32, along with a scaling factor of 32, for parameter-efficient tuning of LLM

Table 3: Performance comparison of connector layer types (linear layer, 2-layer MLP (Mlp2x), and 6-layer MLP (Mlp6x)) on FL fine-tuning on encoder-based VLM undering IID data portions of Fed-SLAKE and Fed-ScienceCap datasets. F-C denotes the connector fine-tuning model, F-L denotes the LLM tuning model. LC denotes joint one-stage connector-LLM tuning and 2stage denotes the sequential fine-tuning of the connector and LLM. The best result is indicated in **bold**, while the second-best result is shown with underline. This performance notation scheme is consistent throughout the paper unless explicitly stated otherwise.

| Mode | Method | Fed-SLAKE | | | Fed-ScienceCap | | |
|------|--------|-----------|--------|--------|----------------|--------|--------|
| | | Linear | Mlp2x | Mlp6x | Linear | Mlp2x | Mlp6x |
| F-C | Central | 0.799 | 0.788 | 0.734 | 7.239/0.879 | 7.361/0.889 | 7.274/0.881 |
| | FedAvg | 0.726 | 0.783 | 0.759 | 7.069/0.867 | 7.283/0.882 | 6.991/0.866 |
| F-L | Central | **0.837** | 0.834 | 0.531 | **7.534**/0.898 | 7.459/0.896 | 5.784/0.833 |
| | FedAvg | 0.787 | 0.806 | 0.794 | 7.498/0.893 | 7.338/0.889 | 5.727/0.832 |
| F-CL | Central | 0.824 | **0.843** | 0.739 | 7.521/**0.899** | **7.550/0.901** | 7.366/0.892 |
| | FedAvg | 0.819 | 0.823 | 0.802 | 7.468/0.896 | 7.521/0.899 | 7.274/0.886 |
| F-2stage | Central | 0.815 | 0.830 | **0.817** | 7.424/0.892 | 7.414/0.894 | **7.491/0.894** |
| | FedAvg | 0.808 | 0.811 | 0.797 | 7.216/0.878 | 7.290/0.883 | 7.226/0.883 |

components. More implementation details are available in Appendix C.

**Baseline FL Algorithms.** We evaluate seven representative FL approaches spanning classical and adaptive heterogeneity optimization paradigms: FedAvg (McMahan et al., 2017), FedProx (Li et al., 2020), FedAvgM (Hsu et al., 2019), FedYogi (Reddi et al., 2020), FedAdam (Reddi et al., 2020) , FFA-LoRA (Sun et al., 2024) and FedSA-LoRA (Guo et al., 2025b). Furthermore, we provide a simplified diagram to illustrate the workflows of thees FL methods, as shown in Fig.4.. To establish performance ceilings, we include a Central baseline trained on aggregated client data.

Due to space limitations, we only present the results on one encoder-based VLM, LLAVA1.5-3B, in the main paper. Additionally, the rank of the LoRA parameters is set to 8 in the main paper. Experimental results for More encoder-based VLMs (Qwen2.5-VL-3B, Qwen2.5-VL-7B), encoder-free VLM (Mono-InternVL-3B), and LoRA with rank 32, are provided in the Appendix D.4.

## 5.2 How to Efficiently Fine-tune Encoder-based VLM in FL?

Our initial exploration focuses on assessing the impact of various popularly utilized connection layers (linear, 2-layer MLP, and 6-layer MLP) along with different fine-tuning strategies on the performance of encoder-based VLM in FL.

**Which connector type—linear, 2-layer MLP, or 6-layer MLP—is most effective for FL fine-tuning of encoder-based VLMs?** As shown in Tab. 3, in the majority of experimental settings (10 out of 16 cases), both the simple linear layer and the 2-layer MLP achieve comparable or superior performance to the more complex 6-layer MLP. This suggests that the added complexity in the connector does not necessarily translate to better performance in FL, a finding consistent with observations in centralized training settings (Lin et al., 2024; Li et al., 2025). We attribute this to the connector's primary role: as both visual encoder and LLM are already well-pretrained, the connector should simply *project* visual features into the LLM's semantic space rather than perform complex transformations. A linear layer or 2-layer MLP is sufficient for alignment without over-engineering, while the 6-layer MLP introduces parameter redundancy that over-processes features, causing deviations from the LLM's semantic space and harming generalization. These drawbacks are further amplified in FL, where a 6-layer MLP may amplify client-specific noise memorization and local overfitting, leading to clear performance degradation compared to the 2-layer alternative. As evidenced by Tab.3 , while the 6-layer MLP occasionally outperforms the 2-layer MLP in centralized scenarios (e.g., F-2stage of Central in Fed-ScienceCap), the 2-layer MLP consistently achieves better results in FL settings. Furthermore, we validate this conclusion under non-IID data distributions (see Appendix D.4.4). Additionally, while the linear layer appears effective, its performance in FL is highly sensitive to parameter initialization (random seeds), leading to unstable training outcomes, especially when each client has limited data (see Fig. 5 in the Appendix D.2). Based on these findings, we conclude that:

> **Takeaway 1**: Compared to a simple linear layer and a complex 6-layer MLP, a 2-layer MLP emerges as the most effective connector regarding performance, computational efficiency, and training stability for fine-tuning VLMs in FL.

Based on previous experimental findings, we employ a 2-layer MLP as the connection layer for all subsequent experiments in this study.

**How should we select FL fine-tuning strategies for different FL tasks in encoder-based VLMs?** In the context of federated fine-tuning in encoder-based VLMs, a key question arises: Which fine-tuning strategy is most effective: (1) connector-only (C, denoted as F-C), (2) LLM-only (L, denoted as F-L), (3) joint connector-LLM tuning (CL, denoted as de-noted as F-CL), or (4) two-stage sequential tuning (C→L, denoted as F-2stage). We systematically evaluate these methods across diverse vision-language tasks under FL constraints.

Table 4: Quantitative comparison of four fine-tuning strategies on multi-type task datasets with IID and non-IID distributions. See Tab.7 (full table) in the Appendix.

| Mode | Method | Fed-SLAKE | | Fed-ScienceCap | | Fed-FGVC | |
|---|---|---|---|---|---|---|---|
| | | IID | Non-IID | IID | Non-IID | IID | Non-IID |
| **F-C** | Central | 0.788 | | 7.361/0.889 | | 0.751 | |
| | FedAvg | 0.783 | 0.775 | 7.285/0.882 | 7.249/0.881 | 0.724 | 0.585 |
| | FedProx | 0.734 | 0.750 | 7.293/0.885 | 7.250/0.881 | 0.726 | 0.586 |
| | FFA-LoRA | 0.787 | 0.776 | 7.281/0.879 | 7.253/0.887 | 0.731 | 0.590 |
| **F-L** | Central | 0.834 | | 7.459/0.896 | | 0.707 | |
| | FedAvg | 0.806 | 0.802 | 7.355/0.890 | 7.342/0.889 | 0.647 | 0.529 |
| | FedProx | 0.800 | 0.780 | 7.331/0.889 | 7.311/0.887 | 0.637 | 0.488 |
| | FFA-LoRA | 0.805 | 0.799 | 7.353/0.887 | 7.339/0.885 | 0.649 | 0.532 |
| **F-CL** | Central | 0.839 | | 7.550/0.901 | | 0.764 | |
| | FedAvg | **0.823** | **0.827** | **7.501/0.898** | **7.476/0.897** | 0.721 | 0.603 |
| | FedProx | 0.816 | 0.796 | 7.500/0.898 | 7.440/**0.897** | 0.718 | 0.548 |
| | FFA-LoRA | 0.821 | 0.819 | 7.473/0.897 | 7.464/**0.897** | 0.722 | 0.602 |
| **F-2stage** | Central | 0.845 | | 7.591/0.908 | | 0.772 | |
| | FedAvg | 0.811 | 0.814 | 7.334/0.884 | 7.281/0.883 | **0.730** | **0.614** |
| | FedProx | 0.773 | 0.785 | 7.262/0.883 | 7.221/0.880 | 0.715 | 0.591 |
| | FFA-LoRA | 0.808 | 0.811 | 7.329/0.880 | 7.278/0.879 | 0.728 | 0.612 |

We begin by examining the impact of fine-tuning either the connector or the LLM across different tasks in FL settings. As detailed in Tab. 4, for the text-dominant tasks (the VQA on Fed-SLAKE and caption generation on Fed-ScienceCap datasets), LLM tuning (F-L) significantly outperforms connector-only tuning (F-C) (e.g., 0.806/0.802 compared to 0.783/0.775 of FedAvg on Fed-SLAKE) , and yields results comparable to full-model tuning (F-CL and F-2stage). Conversely, for vision-centric tasks (fine-grained image classification

tasks on Fed-FGVC), connector tuning (F-C) achieves results comparable to full-model tuning (F-CL and F-2stage), while substantially outperforming LLM-only adaptation (F-L) (e.g., 0.724/0.585 vs. 0.647/0.529 of FedAvg on Fed-FGVC) . This suggests that text-driven tasks benefit from updating linguistic knowledge, whereas vision-centric tasks require refined visual-textual alignment.

> **Takeaway 2**: In federated fine-tuning of VLMs, prioritizing LLM fine-tuning enhances performance in text-centric tasks, such as VQA and caption generation, whereas fine-tuning the connector is more effective for vision-centric tasks like image classification.

Subsequently, we compare full-model fine-tune strategies (F-CL vs. F-2stage). In traditional VLM fine-tuning, it is commonly believed that tuning the connector before the LLM is preferred. However, our findings present an intriguing contrast. As illustrated in Tab. 4, F-2stage generally achieves slightly better performance than F-CL on vision-centric tasks (Fed-FGVC), whereas F-CL demonstrates a marginal advantage on text-centric tasks (Fed-SLAKE). This pattern can be explained as follows: for vision-centric tasks, F-2stage prioritizes connector optimization, which allows for more precise adaptation of visual feature extraction. In contrast, for text-centric tasks, F-CL leverages the dynamic co-adaptation between the connector and the LLM, leading to better integration of textual information. Nevertheless, considering the practical constraints of FL, such as communication costs (see Appendix D.6), F-CL offers a more balanced and communication-efficient strategy, making it preferable for general deployment.

> **Takeaway 3**: For encoder-based VLMs in FL settings, joint fine-tuning works better than sequential training. Specifically, tuning both the connector and LLM together outperforms tuning the connector first, then the LLM, which balances performance gains and computational efficiency.

Based on these experimental findings, we adopt the F-CL as the federated tuning strategy for all subsequent experiments in this study.

**What's the impact of data heterogeneity on federated fine-tuning of encoder-based VLMs?** We now investigate how non-IID (heterogeneous) data distributions influence the performance of encoder-based VLMs on different FL tasks. As shown in Tab. 4, for text-centric tasks (such as VQA and caption generation), there is no significant difference in performance among the various fine-tuning methods under IID and non-IID conditions. However, vision-centric tasks (Fed-FGVC) exhibit a significant performance drop (approximately 20%) under non-IID settings compared to IID baselines. Based on these phenomena, we hypothesize that text-driven tasks inherently benefit from the strong language priors embedded in LLMs. To verify this hypothesis, we conduct two complementary experiments (see Appendix D.5). The results confirm that text-centric tasks maintain functionality under significant visual corruption, and that their resilience to data heterogeneity diminishes substantially when pre-trained language knowledge is absent. Notably, traditional FL optimizers like FedProx and FedYogi fail to address this performance degradation. This conclusion is further supported by experiments on non-IID datasets generated via Dirichlet distributions with varying heterogeneity levels (see Fig.6 in Appendix D.3). These findings highlight the need for new approaches specifically designed to address the challenges posed by data heterogeneity in federated fine-tuning of encoder-based VLMs, particularly for vision-centric tasks.

> **Takeaway 4**: Encoder-based VLMs remain relatively stable on text-centric federated tasks under data heterogeneity, but exhibit significant performance drops for vision-centric tasks in non-IID scenarios. Current FL optimization methods provide limited mitigation in such contexts, underscoring the need for novel solutions specifically designed to tackle data heterogeneity in vision-centric multimodal FL.

### 5.3 HOW DO DIFFERENT VLM ARCHITECTURES RESPOND TO DATA HETEROGENEITY IN FL?

Here, we systematically compare encoder-based and encoder-free VLMs in FL under both IID and non-IID settings, evaluating their performance across tasks with distinct modality dominance: text-centric versus vision-centric. Unlike encoder-based models that separate visual and linguistic components with trainable connectors, encoder-free VLMs operate as unified frameworks without explicit alignment modules (connectors).

Table 5: Performance comparison of different VLM architectures on various single-task datasets with IID and non-IID distributions.

| Mode | Method | Fed-SLAKE | | Fed-ScienceCap | | Fed-FGVC | | Fed-RadGnome | |
|---|---|---|---|---|---|---|---|---|---|
| | | IID | Non-IID | IID | Non-IID | IID | Non-IID | IID | Non-IID |
| | Central | 0.843 | | 7.550/0.901 | | 0.764 | | 0.594 | |
| | FedAvg | **0.823** | **0.827** | 7.501/0.898 | 7.476/0.897 | 0.721 | **0.603** | 0.565 | 0.484 |
| | FedProx | 0.816 | 0.796 | 7.500/0.898 | 7.440/0.897 | 0.718 | 0.548 | 0.535 | 0.462 |
| Encoder-based | FedAdam | 0.777 | 0.774 | 7.282/0.891 | 7.319/0.891 | 0.671 | 0.528 | 0.550 | 0.529 |
| (LLaVA-1.5-3B) | FedAvgM | 0.784 | 0.768 | 7.359/0.893 | 7.351/0.892 | 0.677 | 0.514 | 0.542 | 0.511 |
| | FedYogi | 0.783 | 0.775 | 7.277/0.890 | 7.287/0.890 | 0.675 | 0.511 | 0.556 | **0.536** |
| | FFA-LoRA | 0.821 | 0.819 | 7.473/0.897 | 7.464/0.897 | 0.722 | 0.602 | 0.568 | 0.487 |
| | FedSA-LoRA | 0.817 | 0.806 | **7.524**/0.899 | **7.498**/0.897 | **0.727** | 0.573 | 0.546 | 0.502 |
| | Central | 0.784 | | 7.462/0.899 | | 0.739 | | 0.612 | |
| | FedAvg | 0.777 | 0.761 | 7.470/**0.902** | 7.421/**0.899** | 0.721 | 0.493 | **0.604** | 0.485 |
| | FedProx | 0.769 | 0.734 | 7.456/0.901 | 7.363/0.897 | 0.679 | 0.440 | 0.565 | 0.460 |
| Encoder-free | FedAdam | 0.747 | 0.732 | 7.241/0.894 | 6.850/0.881 | 0.689 | 0.471 | 0.597 | 0.472 |
| (Show-O-3B) | FedAvgM | 0.776 | 0.743 | 7.398/0.899 | 7.402/**0.899** | 0.723 | 0.453 | 0.596 | 0.435 |
| | FedYogi | 0.749 | 0.737 | 7.221/0.893 | 7.267/0.894 | 0.686 | 0.467 | 0.599 | 0.461 |
| | FFA-LoRA | 0.752 | 0.730 | 7.302/0.892 | 7.147/0.886 | 0.675 | 0.481 | 0.566 | 0.474 |
| | FedSA-LoRA | 0.640 | 0.631 | 6.005/0.837 | 5.956/0.819 | 0.477 | 0.304 | 0.515 | 0.483 |

Table 6: Quantitative comparison on two multi-task FL datasets Fed-Nature and Fed-Med datasets. MT-Central refers to centralized training on the centralized multi-task dataset. Note: FedSA-LoRA is optimized for personalized FL scenarios, which explains its limited performance in some global-model-centric evaluations.

| Mode | Method | Fed-Nature | | | | Fed-Med | | |
|---|---|---|---|---|---|---|---|---|
| | | VQA Acc↑ | Caption Generation CIDER↑ ROUGE_L ↑ | Visual Grounding IoU↑ | Classification Acc↑ | VQA Acc↑ | Report Generation CIDER↑ ROUGE_L ↑ | Detection IoU↑ |
| | MT-Central | 0.755 | 0.872/0.358 | 0.405 | 0.913 | 0.674 | 2.101/0.595 | **0.616** |
| | FedAvg | 0.756 | 0.794/0.336 | 0.357 | 0.911 | 0.698 | **2.132**/0.599 | 0.588 |
| Encoder-based | FedProx | 0.711 | 0.807/0.350 | 0.352 | 0.893 | 0.667 | 1.929/0.574 | 0.615 |
| (LLaVA-1.5-3B) | FedAdam | 0.742 | 0.810/0.344 | 0.386 | 0.901 | 0.683 | 2.054/0.589 | 0.567 |
| | FedAvgM | 0.735 | 0.788/0.336 | 0.393 | 0.912 | 0.664 | 1.921/0.576 | 0.592 |
| | FedYogi | 0.744 | 0.784/0.341 | 0.395 | 0.900 | 0.682 | 1.986/0.583 | 0.588 |
| | FFA-LoRA | 0.751 | 0.796/0.341 | 0.362 | 0.912 | 0.702 | 2.098/0.593 | 0.589 |
| | FedSA-LoRA | 0.749 | 0.806/0.349 | 0.371 | **0.924** | **0.707** | 1.819/0.571 | 0.572 |
| | MT-Central | 0.752 | 0.912/0.361 | 0.465 | 0.874 | 0.610 | 1.922/0.575 | 0.581 |
| | FedAvg | **0.781** | 0.930/0.363 | 0.449 | 0.888 | 0.607 | 1.887/0.566 | 0.578 |
| Encoder-free | FedProx | 0.610 | 0.938/0.376 | 0.404 | 0.786 | 0.584 | 1.515/0.538 | 0.532 |
| (Show-O-3B) | FedAdam | 0.739 | **1.090**/0.402 | 0.460 | 0.885 | 0.651 | 1.806/0.564 | 0.579 |
| | FedAvgM | 0.761 | 1.010/0.390 | 0.426 | 0.886 | 0.634 | 1.790/0.555 | 0.604 |
| | FedYogi | 0.742 | 1.072/0.398 | 0.456 | 0.893 | 0.654 | 1.819/0.564 | 0.577 |
| | FFA-LoRA | 0.776 | 0.951/0.382 | **0.468** | 0.898 | 0.612 | 1.883/0.559 | 0.576 |
| | FedSA-LoRA | 0.653 | 0.423/0.236 | 0.0 | 0.768 | 0.418 | 0.402/0.436 | 0.0 |

As shown in Tab. 5, encoder-free VLMs exhibit no significant performance variation on text-centric tasks (Fed-SLAKE VQA and Fed-ScienceCAP caption generation) between IID and non-IID conditions, mirroring the behavior of encoder-based VLMs. This further suggests that text-driven tasks inherently benefit from the linguistic priors of LLMs, regardless of architectural differences. In contrast, for vision-centric tasks (Fed-FGVC classification and Fed-RadGenome detection), both architectures suffer performance degradation under non-IID data. However, the performance drop for the encoder-free model is more pronounced than that of the encoder-based model. This disparity likely arises from the absence of trainable connectors, suggesting that learnable connectors can mitigate some challenges associated with data heterogeneity. Furthermore, consistent with our earlier findings in Sec. 5.2, traditional FL optimizers (e.g., FedProx, FedYogi) demonstrate limited efficacy in mitigating performance degradation for both architectures under non-IID conditions. This emphasizes the necessity for architecture-aware FL optimization strategies specifically tailored to address heterogeneity challenges in vision-centric tasks.

> **Takeaway 5**: Both encoder-based and encoder-free VLMs exhibit robust performance on text-centric FL tasks under non-IID conditions, while vision-centric tasks show pronounced sensitivity to non-IID, with encoder-free VLMs exhibiting larger performance drops. Current FL optimization methods show limited effectiveness in both encoder-free and encoder-based VLMs, calling for novel solutions addressing vision-centric heterogeneity challenges.

### 5.4 How Do Various FL VLM Architectures Perform in Real-world FL Multi-task Scenarios?

Here, we investigate various VLM architectures and FL algorithms on the two multi-task FL datasets (Fed-Nature and Fed-Med). Our evaluation on real-world non-IID multi-task FL benchmarks shows a clear difference from single-task FL results. While single-task FL struggles with vision-centric performance degradation under non-IID data, federated multitask training achieves near-ceiling performance comparable to centralized training across both text- and vision-centric tasks, regardless

of VLM architectures, see Tab. 6. Additionally, while there is no clear winner among the existing FL methods on multi-task learning, the naive FedAvg and FFA-LoRA perform more stably across various tasks compared to other FL-optimized methods. These findings underscore the viability of FL multitask learning as a privacy-preserving alternative to centralized training in real-world multi-task vision-language systems, particularly given the growing prevalence of multitask VLM deployments.

> **Takeaway 6**: Both encoder-based and encoder-free VLMs achieve near-ceiling centralized performance in real-world federated multi-task learning, demonstrating their viability as privacy-preserving alternatives in multitask VLM deployments.

## 6 CONCLUSION

We present FedVLMBench, the first comprehensive benchmark for federated VLM fine-tuning, addressing critical gaps in architectural diversity (encoder-based vs. encoder-free VLMs), task coverage, and multi-task FL scenarios. Through systematic evaluation across 6 datasets, 7 FL algorithms, and 4 fine-tuning strategies, we demonstrate that, in FL fine-tuning of encoder-based VLMs, 2-layer MLP connectors with concurrent connector-LLM tuning are most effective, and we identify task-specific tuning strategies: LLM tuning for text-centric tasks and connector tuning for vision-centric tasks. Notably, our findings reveal that conventional FL optimization methods for vision-centric tasks (e.g., detection) exhibit higher sensitivity to data heterogeneity challenges than text-centric tasks in federated VLM tuning, demanding novel solutions addressing data heterogeneity challenges in vision-centric FL tasks. We hope this work provides foundational support for advancing federated VL systems in real-world applications where data decentralization and task diversity coexist.

**Limitation & Further Work.** Our research primarily focuses on analyzing the performance of different FL algorithms across various VLM architectures, tuning methods, task types, and data distributions. Through comprehensive evaluations, we reveal an urgent need for further research into the performance degradation of vision-centric tasks under non-IID conditions, highlighting the necessity for targeted improvements in this area. However, these experiments primarily focus on data heterogeneity and do not account for the complexities introduced by device or system heterogeneity. Additionally, our current analysis is centered on foundational performance metrics, such as classification accuracy and detection precision. Although we provide an assessment of the communication and computation costs associated with different FL methods and their interplay with VLM architectures in Appendix D.6, this assessment remains insufficient.

Furthermore, several unresolved issues remain, presenting promising directions for future research. Notably, our study is limited to horizontal FL, excluding vertical and hybrid paradigms that could provide valuable insights. In vertical FL, visual modal and linguistic modal data may be distributed across different institutions. How to achieve cross-modal representation alignment without directly sharing modality-specific features has become a core challenge.

We also do not analyze potential privacy leakage or explore privacy-preserving strategies in FL-VLM. For VLM, Cross-modal dependencies create novel attack surfaces—such as using textual prompts to infer visual feature distributions—while conventional privacy-preserving methods prove inadequate: differential privacy impairs fine-tuning utility, and secure multi-party computation cannot scale to VLM-sized models. New mechanisms are needed to balance privacy with utility in this setting.

Morever, fairness remains another important yet underexplored issue in our current benchmark. While we have focused on overall performance metrics, we acknowledge that FL applications in privacy-sensitive domains such as healthcare often face pronounced fairness challenges, such as performance disparities across demographic groups and institutions. The integration of VLMs into federated settings may further amplify such issues due to biases inherited from large-scale pretraining and data distribution shifts among clients. Recent studies(Huang et al., 2024; Zhang et al., 2024a; Wang et al., 2025; Zeng et al., 2024a) have discussed and proposed fairness-enhancing strategies for FL, including fairness-aware optimization and bias mitigation techniques. These efforts generally follow two technical pathways: evaluation-and-optimization based mechanisms for system-level fairness, and data/feature-level interventions using VLM-specific methods like attribute-aware sampling and visual prompt tuning. However, systematic investigation of fairness in FL-VLM settings, and the development of dedicated methods for mitigating bias in federated finetuning of VLMs, remain largely open problems. We hope our benchmark can serve as a foundation for advancing research in fair and equitable FL-VLM systems.

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

## A  THE USE OF LARGE LANGUAGE MODELS (LLMs)

We used Large Language Models (LLMs) to enhance the language and clarity of this manuscript. Their role included rephrasing for readability, correcting grammatical errors, and ensuring consistent terminology. All core scientific contributions, including the proposed methods, experimental design, and results analysis, are original to the authors. The LLMs acted solely as writing assistants and did not influence the research ideas or outcomes presented.

## B  ADDITIONAL INFORMATION ABOUT THE DATASET

### B.1  DISTRIBUTION OF DATASET ATTRIBUTES.

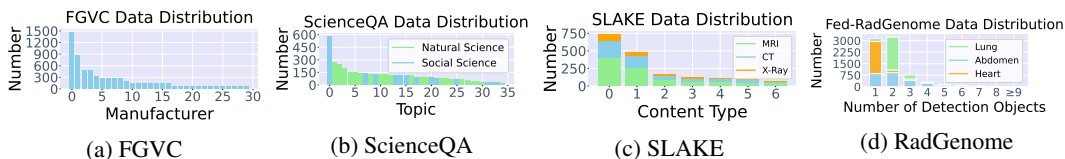

(a) FGVC  (b) ScienceQA  (c) SLAKE  (d) RadGenome

Figure 2: Data distribution of the original datasets. We divide them into clients with different degrees of data heterogeneity based on their labels, such as "Real World Non-IID Distribution" which strictly adheres to the established label categories.

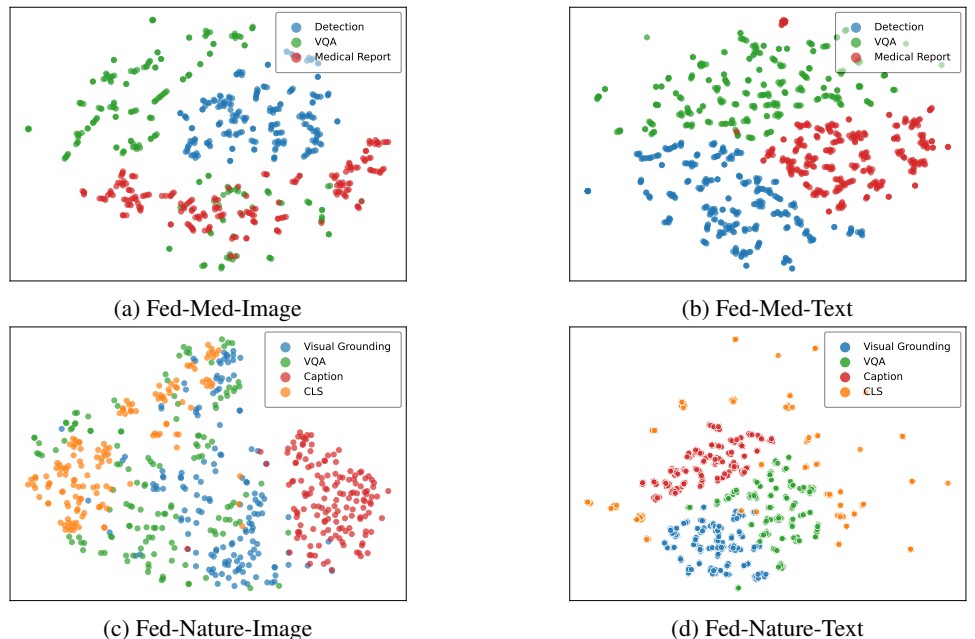

(a) Fed-Med-Image  (b) Fed-Med-Text

(c) Fed-Nature-Image  (d) Fed-Nature-Text

Figure 3: The t-SNE visualization of text and image embeddings from different tasks in Fed-Med and Fed-Nature. Each color denotes one task. We can see clustering phenomenon of one client's data and that clients' data are diverse.

First, as shown in Fig.2, we present the category distribution for four data sources: Fed-FGVC, Fed-ScienceCap, Fed-SLAKE, and Fed-RadGenome. Each dataset contains a diverse array of data points, each with multiple attributes. This richness in diversity makes them ideal for constructing a comprehensive FL dataset.

Next, we also present the t-SNE visualization of text and image embeddings from different tasks in Fed-Nature and Fed-Med datasets in Fig. 3. The images and texts of the two datasets exhibit certain similarities, indicating an overlap in knowledge among different clients. This overlap is conducive to collaborative training, which can improve performance in various tasks.

## C   MORE IMPLEMENT DETAILS

### C.1   TRAINING DETAILS

All experiments are conducted on NVIDIA L40S GPUs. The learning rate is initialized to 1e-4 and optimized using a cosine annealing scheduler to balance convergence speed and stability. The batch size and maximum sequence length for the model architecture are set to 8 and 512, respectively. For the encoder-free model, the codebook size for the image tokens is 8,192. To ensure fairness in evaluation, all additional hyperparameters are adjusted to their default values for benchmark comparison. Hardware-specific optimizations, including mixed precision training and gradient checkpointing, are uniformly applied across all runs to minimize resource discrepancies.

### C.2   METRICS

For the four types of downstream tasks discussed in our benchmark, we employ classic metrics to evaluate the performance of various methods.

1) **Image Classification** primarily employs Accuracy (Acc) to quantify the proportion of correctly predicted labels relative to the total number of samples.

2) **Text Generation** employs CIDEr (Consensus-based Image Description Evaluation), which weights n-gram similarity using TF-IDF to prioritize informative and diverse outputs. Additionally, ROUGE-L evaluates fluency through unigram recall and longest common subsequence alignment.

3) **Visual Question Answering (VQA)** also utilizes Task-Specific Accuracy (Acc), which evaluates exact matches or provides partial credit for semantically equivalent answers.

4) **Detection** relies on Mean Intersection over Union (mIoU), which measures the overlap between the predicted and ground-truth bounding boxes.

### C.3   DATASETS

We provide details on the specific partitions for the following four datasets:

1) **Fed-FGVC:** Utilizing the key attribute "manufacturer" from the FGVC dataset, which consists of 30 categories, we create six distinct partitions.

   - **IID Distribution**: We can allocate up to all 30 categories across a maximum of 30 clients, ensuring that each client receives a representative sample. This results in a balanced distribution among clients.
   - **Real World Non-IID Distribution**: We can also distribute the 30 categories across a maximum of 30 clients by grouping the categories into different sets, assigning each group to a different client. This approach creates imbalanced data distributions, reflecting real-world scenarios where clients have varying amounts of data.
   - **Simulated Non-IID Distribution**: We simulate the non-IID distribution with the Dirichlet Distribution. Specifically, we can similarly allocate the 30 categories across up to 30 clients, varying the alpha parameter to values of 0.01, 0.5, 5, and 100. Each setting generates data sets with different degrees of data heterogeneity, allowing us to analyze the impact of the data distribution on model performance.

   In our experiments, we utilize a total of 5 clients across all distribution methods.

2) **Fed-ScienceCap:** We screen image-description pairs and exclude categories with fewer than 100 samples based on 'category'. We then distribute the remaining 27 image-description pairs among different clients, creating two distinct partitions.

   - **IID Distribution**: We can allocate up to 27 categories across a maximum of 27 clients, evenly distributing different categories to each client to ensure balanced representation.
   - **Real World Non-IID Distribution**: We can also group the 27 categories while maintaining a similar number of data points across the clients, reflecting varying data distributions.

   In our experiments, we utilize a total of 5 clients for both distribution methods.

3) **Fed-SLAKE:** We first exclude question types with fewer than 20 samples, then use uniform and complete partitioning by "modality" to create IID and Non-IID partitions among 3 clients.

- **IID Distribution**: We evenly distribute all data to three clients based on different medical imaging modalities (X-ray, CT, and MRI), resulting in a balanced partition.
- **Real World Non-IID Distribution**: We allocate all images to the three clients according to modality, creating a partition that reflects varying data distributions.

4) **Fed-RadGenome:** We first extract 2D images from the 3D volume segmentation mask in the RadGenome-Chest CT dataset. Specifically, we extract two axial slices at normalized heights of 35% (z=0.35) and 50% (z=0.5) along the superior-inferior axis, generating two 2D images from the same case. We select three organs—lung, abdomen, and heart—as our detection targets, each potentially containing multiple detection targets. For each irregularly shaped detection target, we generate axis-aligned bounding boxes based on spatial extremal points.

- **IID Distribution**: We evenly distribute all data to three clients based on organ type, resulting in a balanced IID partition that ensures that each client has a similar representation of the data.
- **Real World Non-IID Distribution**: We allocate the data to the three clients according to specific organ categories, creating a Non-IID partition that reflects varying data distributions among clients.

**Extension to Diverse Client Scales**: Beyond the specific client configurations presented above, our FedVLMBench is explicitly designed to support flexible scaling in the number of clients. Specifically, our released codebase provides built-in utilities (e.g., Dirichlet distribution-based partitioning) that enable researchers to simulate federated scenarios with any number of clients, from just a few to hundreds or more. This ensures the comprehensive studies on scalability and data heterogeneity.

## C.4 BASELINE

We evaluate six representative FL approaches spanning classical and adaptive heterogeneity optimization paradigms: FedAvg (McMahan et al., 2017), FedProx (Li et al., 2020), FedAvgM (Hsu et al., 2019), FedYogi (Reddi et al., 2020), FedAdam (Reddi et al., 2020), FFA-LoRA (Sun et al., 2024) and FedSA-LoRA (Guo et al., 2025b). Furthermore, we provide a simplified diagram to illustrate the workflows of thees key FL methods, as shown in Fig.4.

1) **FedAvg**: A foundational method in federated learning (FL) that performs local multi-round training on clients before uploading model parameters. It aggregates these parameters through weighted averaging, significantly reducing communication rounds and ensuring data privacy.

2) **FedProx**: This method builds upon FedAvg by introducing a proximal regularization term to limit the deviation of local models from the global model. The regularization strength is set to $\mu = 0.01$, which helps alleviate convergence issues caused by data and device heterogeneity, thereby enhancing model robustness in non-IID data distributions.

3) **FedAvgM**: Enhancing the parameter aggregation process of FedAvg, FedAvgM incorporates a momentum mechanism that introduces a momentum term during global model updates. This approach smooths out historical gradient directions, accelerating convergence and stabilizing updates, particularly in complex client data distributions. The momentum parameters are set as follows: $\beta_1 = 0.9$ for the first moment and $\beta_2 = 0.99$ for the second moment.

4) **FedYogi**: An adaptive FL algorithm based on the Yogi optimizer, which dynamically adjusts learning rates to effectively tackle non-convex optimization problems. This method estimates second-order moments of gradients, improving global update directions in heterogeneous data scenarios. The adaptation is controlled by a parameter $\tau = 0.001$, which influences the learning rate adjustments.

5) **FedAdam**: This method integrates the momentum mechanism and adaptive learning rates of the Adam optimizer into the federated framework. By leveraging weighted first and second moments of gradients, it achieves faster convergence and greater stability compared to FedAvg. The parameters include $\beta_1 = 0.9$ and $\beta_2 = 0.99$ for the momentum coefficients, facilitating effective global parameter updates.

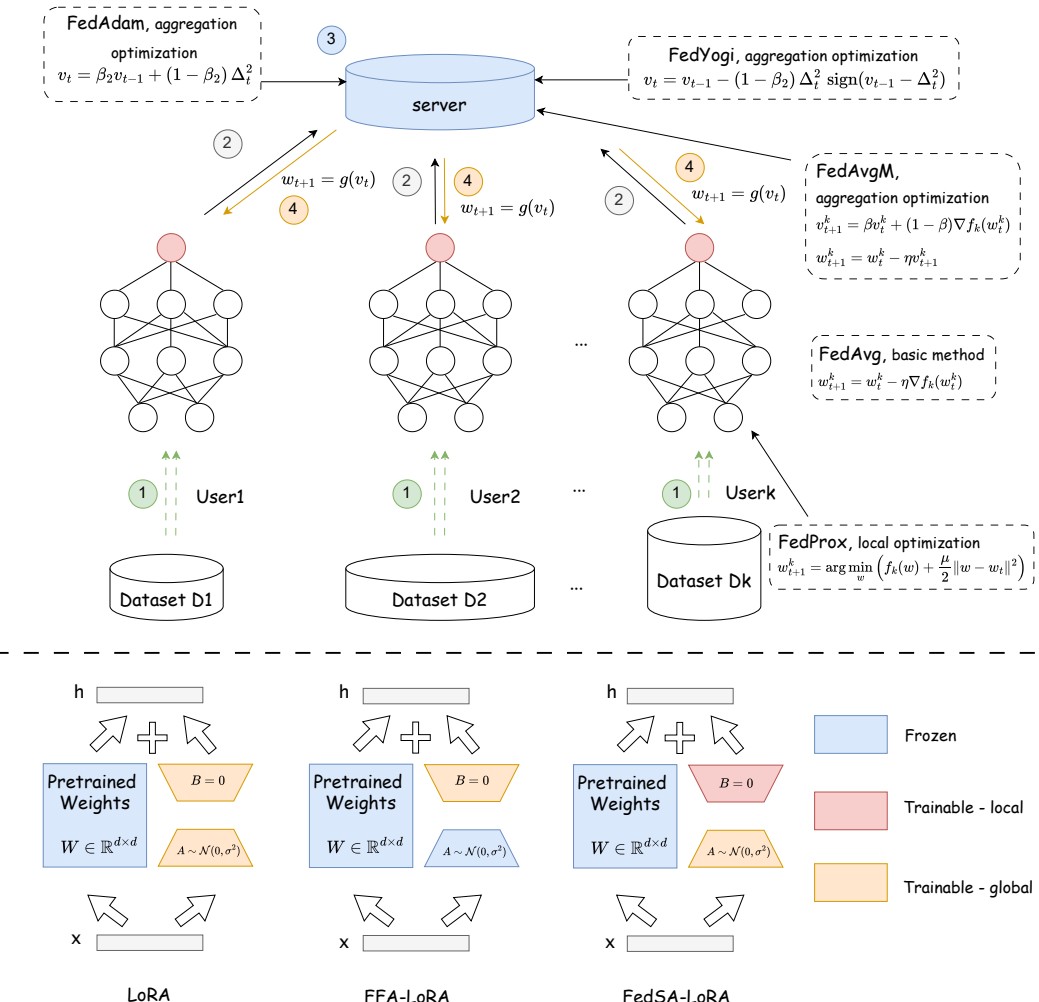

Figure 4: **Up**: Basic FL process can be divided into 4 steps. (1) The client updates the model independently using its local data. (2) The client sends only the model updates (parameters or gradients) produced during local training back to the server. (3) The server aggregates the updates uploaded by the clients. (4) Server sends back updates to the client. Different from basic FedAvg, two main approaches have been proposed to tackle data heterogeneity in FL: (A) Regularizing local training to mitigate the deviation between local and global objectives (modify process 1), like FedProx. (B) Designing more efficient global aggregation strategies (modify process 3), such as FedAdam, FedYogi, and FedAVgM. **Down**: In LoRA, both the A and B matrices are trainable and are shared with the server for aggregation. In FFA-LoRA, only the B matrices are trainable and shared with the server for aggregation, while the A matrices are fixed after initialization. In FedSA-LoRA, both A and B matrices are trainable, but only the A matrices are shared with the server for aggregation, whereas the B matrices remain local to each client.

6) **FFA-LoRA**: FFA-LoRA freezes the non-zero initialized low-rank matrices and only updates and aggregates the zero-initialized matrices. This approach addresses the issue of mismatches that arise when two low-rank matrices are jointly updated locally and aggregated globally.

7) **FedSA-LoRA**: FedSA-LoRA only shares matrix A with the central server for aggregation. Matrix B, however, remains local to the client and does not participate in server-side aggregation, thus preserving personalized knowledge for each client. Following the authors' suggestion, we average matrix B after training and then test it. Notably, FedSA-LoRA is optimized for personalized FL scenarios, which explains its limited performance in some global-model-centric evaluations.

# D ADDITIONAL EXPERIMENTAL RESULTS

## D.1 FULL EXPERIMENTAL RESULTS

As shown in Tab. 7, we provide the complete versions of Tab. 4 in our main paper.

Table 7: Quantitative comparison of four fine-tuning strategies on multi-type task datasets with IID and non-IID distributions.

| Mode | Method | Fed-SLAKE | | Fed-ScienceCap | | Fed-FGVC | |
|---|---|---|---|---|---|---|---|
| | | IID | Non-IID | IID | Non-IID | IID | Non-IID |
| F-C | Central | 0.788 | | 7.361/0.889 | | 0.751 | |
| | FedAvg | 0.783 | 0.775 | 7.285/0.882 | 7.249/0.881 | 0.724 | 0.585 |
| | FedProx | 0.734 | 0.750 | 7.293/0.885 | 7.250/0.881 | 0.726 | 0.586 |
| | FedAdam | 0.741 | 0.735 | 7.127/0.876 | 7.137/0.876 | 0.694 | 0.522 |
| | FedAvgM | 0.754 | 0.747 | 7.252/0.880 | 7.238/0.881 | 0.696 | 0.510 |
| | FedYogi | 0.745 | 0.736 | 7.125/0.877 | 7.104/0.874 | 0.695 | 0.511 |
| | FFA-LoRA | 0.787 | 0.776 | 7.281/0.879 | 7.253/0.887 | 0.731 | 0.590 |
| F-L | Central | 0.834 | | 7.459/0.896 | | 0.707 | |
| | FedAvg | 0.806 | 0.802 | 7.355/0.890 | 7.342/0.889 | 0.647 | 0.529 |
| | FedProx | 0.800 | 0.780 | 7.331/0.889 | 7.311/0.887 | 0.637 | 0.488 |
| | FedAdam | 0.783 | 0.771 | 7.194/0.885 | 7.125/0.881 | 0.627 | 0.460 |
| | FedAvgM | 0.789 | 0.786 | 7.287/0.890 | 7.305/0.890 | 0.602 | 0.469 |
| | FedYogi | 0.782 | 0.769 | 7.153/0.884 | 7.123/0.881 | 0.623 | 0.467 |
| | FFA-LoRA | 0.805 | 0.799 | 7.353/0.887 | 7.339/0.885 | 0.649 | 0.532 |
| F-CL | Central | 0.839 | | 7.550/0.901 | | 0.764 | |
| | FedAvg | **0.823** | **0.827** | **7.501/0.898** | **7.476/0.897** | 0.721 | 0.603 |
| | FedProx | 0.816 | 0.796 | 7.500/**0.898** | 7.440/**0.897** | 0.718 | 0.548 |
| | FedAdam | 0.777 | 0.774 | 7.282/0.891 | 7.319/0.891 | 0.671 | 0.528 |
| | FedAvgM | 0.784 | 0.768 | 7.359/0.893 | 7.351/0.892 | 0.677 | 0.514 |
| | FedYogi | 0.783 | 0.774 | 7.277/0.890 | 7.287/0.890 | 0.675 | 0.511 |
| | FFA-LoRA | 0.821 | 0.819 | 7.473/0.897 | 7.464/**0.897** | 0.722 | 0.602 |
| F-2stage | Central | 0.845 | | 7.591/0.908 | | 0.772 | |
| | FedAvg | 0.811 | 0.814 | 7.334/0.884 | 7.281/0.883 | **0.730** | **0.614** |
| | FedProx | 0.773 | 0.785 | 7.262/0.883 | 7.221/0.880 | 0.715 | 0.591 |
| | FedAdam | 0.782 | 0.777 | 7.315/0.887 | 7.315/0.887 | 0.713 | 0.539 |
| | FedAvgM | 0.793 | 0.794 | 7.369/0.889 | 7.380/0.889 | 0.708 | 0.565 |
| | FedYogi | 0.785 | 0.782 | 7.310/0.886 | 7.310/0.886 | 0.717 | 0.561 |
| | FFA-LoRA | 0.808 | 0.811 | 7.329/0.880 | 7.278/0.879 | 0.728 | 0.612 |

Table 8: Quantitative comparison of Qwen2.5-VL-3B on two representative single-task datasets with IID and non-IID distributions.

| Method | Fed-SLAKE | | Fed-FGVC | |
|---|---|---|---|---|
| | IID | Non-IID | IID | Non-IID |
| Central | 0.804 | | 0.871 | |
| FedAvg | 0.771 | 0.756 | 0.850 | 0.812 |
| FedProx | 0.772 | 0.761 | 0.843 | 0.765 |
| FedAdam | 0.760 | 0.754 | 0.831 | 0.740 |
| FedAvgM | 0.759 | 0.752 | 0.834 | 0.758 |
| FedYogi | 0.755 | 0.752 | 0.830 | 0.703 |
| FFA-LoRA | 0.769 | 0.757 | 0.847 | 0.801 |

## D.2 EXPERIMENTS ON THE STABILITY OF CONNECTOR TRAINING

As shown in Fig.5, we show the results of different connector types in two scenarios: FL and centralized training. It can be seen that the performance of the linear layer in FL fluctuates significantly more than that in centralized training, which indicates that the linear layer is unstable in the FL scenario with limited data (each client in the FL scenario only has a portion of the data in the centralized training mode).

Table 9: Quantitative comparison of Qwen2.5-VL-3B on Fed-Nature dataset. MT-Central refers to centralized training on the centralized multi-task dataset.

| Method | VQA Acc↑ | Caption Generation CIDER↑ ROUGE_L ↑ | Visual Grounding IoU ↑ | Classification Acc↑ |
|---|---|---|---|---|
| MT-Central | 0.784 | 1.440 / 0.446 | 0.681 | 0.975 |
| FedAvg | 0.777 | 1.416 / 0.440 | 0.642 | 0.973 |
| FedProx | 0.779 | 1.483 / 0.452 | 0.630 | 0.971 |
| FedAdam | 0.799 | 1.495 / 0.452 | 0.619 | 0.979 |
| FedAvgM | 0.774 | 1.353 / 0.433 | 0.621 | 0.970 |
| FedYogi | 0.793 | 1.517 / 0.453 | 0.616 | 0.977 |
| FFA-LoRA | 0.781 | 1.439 / 0.443 | 0.639 | 0.975 |

Table 10: Quantitative comparison of Mono-InternVL-3B on two representative single-task datasets with IID and non-IID distributions.

| Method | Fed-SLAKE | | Fed-FGVC | |
|---|---|---|---|---|
| | IID | Non-IID | IID | Non-IID |
| Central | 0.804 | | 0.561 | |
| FedAvg | 0.621 | 0.615 | 0.504 | 0.401 |
| FFA-LoRA | 0.622 | 0.616 | 0.514 | 0.402 |

### D.3 EXPERIMENTS ON VARYING DEGREES OF DATA HETEROGENEITY

Here, we explore the performance of different FL algorithms across various levels of data heterogeneity. As illustrated in Fig.6, traditional FL optimizers such as FedProx and FedYogi do not effectively mitigate this performance degradation.

### D.4 EXPERIMENT ON THE GENERALIZABILITY OF TAKEAWAY TIPS

Here, we further verify the generalization ability of the proposed takeaway tips in different settings, including the advanced VLMs, different LoRA parameters and varying random seeds.

#### D.4.1 EXPERIMENTS ON ADVANCED MODEL.

In order to further verify whether the proposed takewawy tips are effective for the Advanced model, we conduct quantitative experiments for different FL methods on text-centric dataset (Fed-SLAKE), vision-centric dataset (Fed-FGVC), and multi-task FL dataset(Fed-Nature) for Qwen2.5-VL-3B, Qwen2.5-VL-7B and Mono-InternVL-3B.

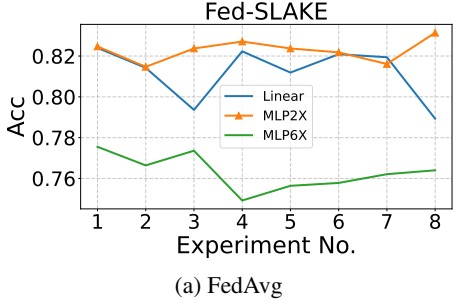
(a) FedAvg

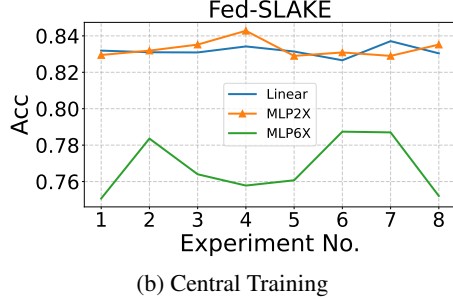
(b) Central Training

Figure 5: Performance of different connection layers across multiple random experiments on the Fed-SLAKE dataset. We conduct eight evaluations using FedAvg and Central training respectively.(mentioned in Sec. 5.2)

Table 11: Quantitative comparison of Mono-InternVL-3B on Fed-Nature dataset. MT-Central refers to centralized training on the centralized multi-task dataset.

| Method | VQA Acc↑ | Caption Generation CIDER↑ ROUGE_L ↑ | Visual Grounding IoU ↑ | Classification Acc↑ |
|---|---|---|---|---|
| MT-Central | 0.686 | 1.020/0.398 | 0.326 | 0.911 |
| FedAvg | 0.711 | 1.080/0.406 | 0.337 | 0.930 |
| FFA-LoRA | 0.663 | 1.037/0.398 | 0.312 | 0.906 |

Table 12: Quantitative comparison of Qwen2.5-VL-7B on two representative single-task datasets with IID and non-IID distributions.

| Method | Fed-SLAKE IID | Fed-SLAKE Non-IID | Fed-FGVC IID | Fed-FGVC Non-IID |
|---|---|---|---|---|
| Central | 0.808 | | 0.913 | |
| FedAvg | 0.781 | 0.785 | 0.885 | 0.851 |
| FFA-LoRA | 0.746 | 0.750 | 0.869 | 0.796 |

Firstly, as shown in Tab. 8, for the text-centric task (Fed-SLAKE), the performance differences between the five FL methods on Qwen2.5-VL-3B are not significant under IID and non-IID conditions. However, for the vision-centric task (Fed-FGVC), both methods show a decrease in performance under non-IID data, with FedAvg outperforming the other methods in this setting. This observation aligns with Takeaway 5. Moreover, we report the performance of different FL methods on the Fed-Nature dataset in Tab. 9. It can be seen that federated multi-task training is close to the performance of centralized training in VQA, caption generation, and classification tasks, which is consistent with Takeaway 6.

Next, as illustrated in Tab. 10, we evaluate the performance on Fed-SLAKE and Fed-FGVC datasets under both IID and non-IID settings with Mono-InternVL-3B. For the text-centric task (Fed-SLAKE), both FL methods perform similarly, regardless of the data distribution (IID or non-IID). In contrast, the vision-centric task reveals a clear performance decline under non-IID conditions. This finding validates the conclusion presented in Takeaway 5. On the multi-task Fed-Nature dataset (Tab. 11), federated multi-modal learning achieves results comparable to centralized training, particularly for FedAvg, in line with Takeaway 6.

Finally, we also repeat the same experiments with the larger Qwen2.5-VL-7B model. Importantly, our experiments with this larger Qwen2.5-VL-7B model (see Tab.12 and Tab. 13,) further confirm that both Takeaway 5 and Takeaway 6 generalize well to large-scale encoder-based VLMs.

In summary, our extended experiments across multiple model scales and architectures—including encoder-based VLMs (Qwen2.5-VL-3B, Qwen2.5-VL-7B) and encoder-free VLMs (Show-O-3B and Mono-InternVL-3B)—consistently validate our core findings. These results strengthen the general applicability of the takeaways proposed in our study.

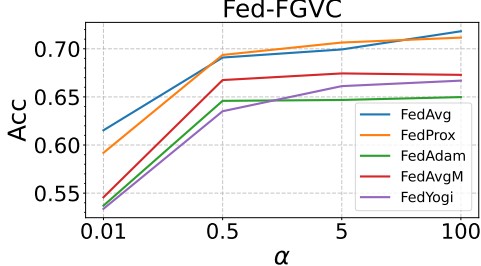

Figure 6: Performance of different FL algorithms at various levels of data heterogeneity on the Fed-FGVC dataset. $\alpha$ is the parameter of the Dirichlet distribution.(mentioned in Sec. 5.2)

Table 13: Quantitative comparison of Qwen2.5-VL-7B on Fed-Nature dataset. MT-Central refers to centralized training on the centralized multi-task dataset.

| Method | VQA Acc↑ | Caption Generation CIDER↑ ROUGE_L ↑ | Visual Grounding IoU ↑ | Classification Acc↑ |
|---|---|---|---|---|
| MT-Central | 0.823 | 1.545/0.463 | 0.675 | 0.982 |
| FedAvg | 0.820 | 1.542/0.462 | 0.647 | 0.981 |
| FFA-LoRA | 0.791 | 1.537/0.457 | 0.598 | 0.976 |

Table 14: Quantitative comparison of training stability on Fed-SLAKE and Fed-FGVC datasets. Each cell contains the mean and variance of three training results.

| Mode | Method | Fed-SLAKE | | Fed-FGVC | |
|---|---|---|---|---|---|
| | | IID | Non-IID | IID | Non-IID |
| Encoder-based | Fedavg | 0.819(4e-5) | 0.822(5e-5) | 0.725(4e-5) | 0.507(2e-5) |
| Encoder-free | Fedavg | 0.773(4e-5) | 0.757(2e-5) | 0.716(5e-5) | 0.492(2e-6) |

### D.4.2 EXPERIMENTS ON RANDOM SEEDS.

Next, to further investigate whether the proposed takeaway tips are influenced by the random seed, we conduct parameter initialization sensitivity experiments on the Fed-SLAKE, Fed-FGVC, and Fed-Nature datasets. Specifically, we follow the same settings as in Tab. 4 and Tab. 5, using three different random seeds for these experiments. We then calculate the mean and variance of all results to assess consistency.

Firstly, as illustrated in Tab. 14, we measure FedAvg's performance on Fed-SLAKE and Fed-FGVC datasets. For the text-centric task (Fed-SLAKE), there is no significant difference in FedAvg's performance between IID and Non-IID distribution. However, for the vision-centric task (Fed-FGVC), we observe a considerable drop in performance under Non-IID conditions, regardless of VLM architectures. This finding supports Takeaway 5 from our paper.

Secondly, we compare the performance between centralized training (MT-Central) and FedAvg in multi-task federated fine-tuning on Fed-Nature dataset. As shown in Tab. 15, FedAvg's performance in four tasks closely matches that of centralized training, which aligns with Takeaway 6 in our paper.

### D.4.3 EXPERIMENTS ON DIFFERENT LoRA PARAMETERS.

Furthermore, we utilize more LoRA parameters (rank=32, alpha=256) to verify the representative conclusions of Takeaway 1, 5 and 6 based on both VLMs, following the same settings as in Tab. 3, Tab. 4, and Tab. 5 of the main paper.

First, for encoder-based VLM, we present the performance of different connectors based on the Fed-SLAKE dataset for the F-CL mode (joint fine-tuning of the connector and LLM) in Tab. 16. The results indicate that the 2-layer MLP remains the best choice, consistent with Takeaway 1 from our paper.

Next, as shown in Tab. 17, we evaluate FedAvg's performance on the Fed-SLAKE and Fed-FGVC datasets. For the text-centric task (Fed-SLAKE), there is no significant difference in FedAvg's performance under IID and non-IID conditions. However, for the vision-centric task (Fed-FGVC), performance drops significantly under non-IID conditions, regardless of VLM architectures. This observation aligns with Takeaway 5 from our paper.

Finally, as illustrated in Tab. 18, we present the performance differences between centralized training (MT-Central) and FedAvg in multi-task federated fine-tuning on the Fed-Nature dataset. Overall, except for the detection task, FedAvg's performance in the other tasks is comparable to that of centralized training, consistent with Takeaway 6 from our paper.

These results confirm that our main conclusions remain valid even with the revised LoRA parameters.

Table 15: Quantitative comparison of training stability on Fed-Nature dataset. Each cell contains the mean and variance of three training results. MT-Central refers to centralized training on the centralized multi-task dataset.

| Mode | Method | VQA Acc↑ | Caption Generation CIDER↑ ROUGE_L ↑ | Visual Grounding IoU ↑ | Classification Acc↑ |
|---|---|---|---|---|---|
| Encoder -based | MT-Central | 0.750(5e-5) | 0.879(8e-6) / 0.359(4e-5) | 0.402(2e-5) | 0.909(3e-5) |
| | Fedavg | 0.747(6e-5) | 0.796(4e-5) / 0.338(3e-5) | 0.361(7e-5) | 0.909(4e-5) |
| Encoder -free | MT-Central | 0.755(2e-5) | 0.911(5e-6) / 0.365(2e-5) | 0.461(4e-5) | 0.876(2e-5) |
| | Fedavg | 0.772(2e-4) | 0.921(2e-4) / 0.362(1e-6) | 0.452(3e-4) | 0.892(2e-5) |

Table 16: Quantitative comparison of connector layer types for FL fine-tuning on an encoder-based VLM under IID data portions of Fed-SLAKE (LoRA parameters: rank=32, alpha=256).

| Method | Linear | Mlp2x | Mlp6x |
|---|---|---|---|
| Fedavg | 0.792 | 0.803 | 0.761 |

### D.4.4 EXPERIMENTS ON VARIOUS FINE-TUNING STRATEGIES FOR CROSS-TASK DATASET.

To further verify the generality of Takeaway tips 1, we present the performance of different connector for the encoder-based VLM undering Non-IID data portions of Fed-SLAKE and Fed-ScienceCap datasets. As shown in Tab.in Tab. 19, which follows the same setup under Non-IID distributions as Tab.3, the MLP2x connector remains the most efficient choice across different tasks and fine-tuning strategies, even under data heterogeneity. This conclusion is consistent with Takeaway 1 from our paper.

Moreover, to verify the generality of Takeaway tip 2 and 3, we present the performance of four fine-tuning strategies in Tab. 20 for the encoder-based VLM on the representative cross-task dataset (Fed-Nature). The findings indicate that fine-tuning only the connector (F-C) significantly enhances performance in classification tasks by effectively aligning the image space with the feature space. Conversely, fine-tuning solely the LLM (F-L) proves beneficial for text-related tasks, such as Visual Question Answering (VQA) and Caption Generation. Notably, the performance of joint fine-tuning (F-CL) surpasses that of the two-stage fine-tuning (F-2stage). These conclusions are consistent with Takeaway 2 in Sec. 5.2.

### D.5 EXPERIMENT ON VERIFYING THE CAUSES OF ROBUSTNESS IN TEXT-CENTRIC TASKS.

To verify our key findings and the underlying mechanisms, especially the hypothesis that the robustness of text-centric tasks stems from the linguistic priors of pre-trained LLMs, we conduct two complementary experiments.

First, we explore the impact of visual inputs on performance in text-centric task and vision-centric task. Specifically, we evaluate the performance of both vision- (Fed-FGVC) and text-centric tasks (Fed-SLAKE) under varying degrees of image occlusion (0%, 20%, 50%, 80%, and 99%). As shown in Tab.21, the vision-centric task suffers a sharp performance drop (84.5%) even under mild occlusion (20%), while the text-centric task declined by only 38.2% even under near-total occlusion (99%). This strongly suggests that text-driven tasks can rely on pre-trained knowledge to compensate for missing visual information, thereby reducing their sensitivity to data heterogeneity.

Moreover, to further verify whether pre-trained language priors are responsible for the robustness of text-centric tasks, we compare the performance of text-centric tasks under both IID and non-IID settings using either pre-trained or randomly initialized LLMs in Tab.22. As expected, the randomly initialized models suffer much greater performance degradation under non-IID data (7.19% degradation), highlighting the importance of pre-trained linguistic knowledge in mitigating the negative effects of data heterogeneity.

Table 17: Quantitative comparison of different tasks under IID and Non-IID distributions on Fed-SLAKE and Fed-FGVC dataset(LoRA parameters: rank=32, alpha=256).

| Mode | Method | Fed-SLAKE | | Fed-FGVC | |
|---|---|---|---|---|---|
| | | IID | Non-IID | IID | Non-IID |
| Encoder-based | Fedavg | 0.803 | 0.801 | 0.728 | 0.622 |
| Encoder-free | Fedavg | 0.782 | 0.773 | 0.724 | 0.505 |

Table 18: Quantitative comparison on the Fed-Nature dataset(LoRA parameters: rank=32, alpha=256).

| Mode | Method | VQA Acc↑ | Caption Generation CIDER↑ ROUGE_L ↑ | Visual Grounding IoU ↑ | Classification Acc↑ |
|---|---|---|---|---|---|
| Encoder-based | MT-Central | 0.745 | 0.893/0.358 | 0.398 | 0.914 |
| | Fedavg | 0.736 | 0.813/0.343 | 0.339 | 0.909 |
| Encoder-free | MT-Central | 0.753 | 0.925/0.362 | 0.478 | 0.882 |
| | Fedavg | 0.785 | 0.940/0.379 | 0.442 | 0.892 |

## D.6 EXPERIMENT ON COMMUNICATION AND COMPUTATIONAL COSTS.

Here, considering that the number of parameters in VLMs is significantly larger than in conventional CNNs and other network architectures, we explore the impact of this feature on federated fine-tuning.

First, to compare the communication cost of joint fine-tuning (F-CL) and sequential fine-tuning (F-2stage) strategy, we report the communication rounds and model parameters transmission (GB) required to achieve the target performance on the Fed-SLAKE and Fed-FGVC datasets with the IID data distribution based on encoder-based model (LLaVA-1.5-3B). As shown in Tab. 23, on the Fed-SLAKE dataset, F-CL achieves target accuracy 0.8 in 15 communication rounds with 6.69 GB parameter transmission, whereas F-2stage requires 41 communication rounds in all and 9.31 GB transmission to reach 0.8 accuracy. A similar phenomenon is observed on Fed-FGVC dataset. This makes the simpler F-CL a superior choice from a practical systems perspective.

Next, we assess the communication costs of various FL methods on the Fed-SLAKE dataset for both IID and non-IID distributions (see Tab. 24). Our results indicate that, regardless of the VLM architecture, FedAvg consistently converges faster than other methods under IID and Non-IID distribution. Additionally, the encoder-based VLM demonstrates significantly faster convergence compared to the encoder-free VLM. We attribute this to the learnable connector's ability to better align image and text tokens, facilitating more efficient optimization. These findings align with Takeaway 5 from our main paper, which notes that encoder-free VLMs experience larger performance drops than their encoder-based counterparts.

Moreover, we measure the average time required for different FL methods to complete a single round of communication on the Fed-Nature dataset, which includes both client training and server-side parameter aggregation. To ensure fairness, we use the same LoRa parameters across all experiments, maintaining a consistent number of parameters optimized by each FL method. The computational loss primarily arises from variations in their optimization approaches. As shown in Tab. 25, the basic method, FedAvg, which directly aggregates parameters from different clients, employs a straightforward and efficient computational strategy. FFA-LoRA, as an improved algorithm of FedAvg, only trains and transmits the B matrix in LoRA, further reducing the computational cost. In contrast, FedProx regularizes model parameters during optimization, resulting in a significant increase in computational cost.

## D.7 EXPERIMENT ON SYSTEM HETEROGENEITY.

Considering that system heterogeneity (heterogeneous environments and varying numbers of clients) is a key practical challenge in FL, we also conduct preliminary experiments to explore the system heterogeneity of FL-VLM.

Table 19: Performance comparison of connector layer types (linear layer, 2-layer MLP (Mlp2x), and 6-layer MLP (Mlp6x)) on FL fine-tuning on encoder-based VLM undering non-IID data portions of Fed-SLAKE and Fed-ScienceCap datasets.

| Mode | Method | Fed-SLAKE | | | Fed-ScienceCap | | |
|---|---|---|---|---|---|---|---|
| | | Linear | Mlp2x | Mlp6x | Linear | Mlp2x | Mlp6x |
| F-C | FedAvg | 0.712 | 0.775 | 0.744 | 7.037/0.863 | 7.249/0.881 | 6.998/0.866 |
| F-L | FedAvg | 0.775 | 0.802 | 0.785 | 7.338/0.888 | 7.342/0.889 | 7.182/0.885 |
| F-CL | FedAvg | 0.821 | 0.827 | 0.805 | 7.404/0.893 | 7.476/0.897 | 7.150/0.882 |
| F-2stage | FedAvg | 0.815 | 0.814 | 0.810 | 7.140/0.874 | 7.281/0.883 | 7.258/0.880 |

Table 20: Quantitative comparison of four fine-tuning strategies of encoder-based VLMs on the Fed-Nature dataset.

| Method | VQA Acc↑ | Caption Generation CIDER↑ ROUGE_L ↑ | Visual Grounding IoU ↑ | Classification Acc↑ |
|---|---|---|---|---|
| F-C | 0.593 | 0.534 / 0.269 | 0.200 | 0.869 |
| F-L | 0.690 | 0.677 / 0.326 | 0.321 | 0.892 |
| F-CL | 0.756 | 0.794 / 0.336 | 0.357 | 0.911 |
| F-2stage | 0.677 | 0.586 / 0.289 | 0.240 | 0.890 |

Specifically, we conduct additional experiments on two representative datasets: Fed-SLAKE (text-centric) and Fed-FGVC (vision-centric). For both datasets, we partition the data into 50 clients in an IID manner, ensuring that each label was evenly distributed across clients. Following prior works (Lai et al., 2022; Chen et al., 2023), we simulate heterogeneous FL environments by introducing "stragglers," defined as clients unable to complete local training within a set communication deadline. The straggler ratio quantifies the proportion of selected clients that fail to participate in model aggregation per round. For example, a straggler ratio of 0% indicates a fully homogeneous environment where all selected clients successfully complete their training, while a ratio of 10% means that, on average, 10% of selected clients fail to finish training in each round.

As shown in Tab.26, for the text-driven Fed-SLAKE dataset, we observe that increasing the straggler ratio from 0.1 to 0.7 only modestly increases the communication cost (e.g., 27/25×). In contrast, for the vision-centric Fed-FGVC dataset, the communication cost rises more sharply (e.g., 38/25×). This trend is consistent with our previous observations regarding non-IID distributions: text-driven tasks tend to benefit from the inherent linguistic priors of large language models, making them more robust to system heterogeneity; whereas vision-centric tasks, which rely more heavily on fine-tuning the visual encoder and connector, show greater vulnerability to uneven client participation.

These experiments demonstrate that FedVLMBench is readily extensible to more realistic FL scenarios. A comprehensive study of the combined effects of system and data heterogeneity is a promising direction for future work, and our benchmark provides a solid foundation for such research.

Table 21: The Impact of different image occlusion rate on Show-O's Performance in text-centric task (Fed-SLAKE dataset) and vision-centric task (Fed-FGVC dataset).

| Occlusion rate | Fed-SLAKE | Fed-FGVC |
|---|---|---|
| 0 | 0.784 | 0.739 |
| 20% | 0.657(-16.2%) | 0.114(-84.5%) |
| 50% | 0.545(-30.4%) | 0.049(-93.3%) |
| 80% | 0.509(-35.1%) | 0.036(-95.1%) |
| 99% | 0.484(-38.2%) | 0.019(-97.4%) |

Table 22: The impact of LLM pre-training on text-centric tasks (Fed-SLAKE dataset).

| Method | IID | Non-IID |
|---|---|---|
| Pretrained Show-O | 0.777 | 0.761 |
| Random init Show-O | 0.681 | 0.632 |

Table 23: Total transmitted message size (number of communication rounds and model parameters transmission (GB) in all) required by FedAvg to reach the target performance (accuracy 0.8 for Fed-SLAKE and 0.7 for Fed-FGVC) based on encoder-based VLM (LLaVA-1.5-3B).

| Mode | Method | Fed-SLAKE(0.8) | Fed-FGVC(0.7) |
|---|---|---|---|
| F-CL | FedAvg | 15 (6.69) | 33 (14.72) |
| F-2stage | FedAvg | 41 (9.31) | 72 (16.11) |

Table 24: The number of communication rounds required for each method to reach the predefined target performance (ACC = 0.7) on the Fed-SLAKE dataset.

| Method | Encoder-based | | Encoder-free | |
|---|---|---|---|---|
| | IID | Non-IID | IID | Non-IID |
| Fedavg | 4 | 8 | 8 | 15 |
| Fedprox | 4 | 10 | 10 | 20 |
| FedAdam | 14 | 15 | 29 | 40 |
| FedAvgM | 11 | 21 | 25 | 32 |
| FedYogi | 14 | 16 | 29 | 41 |
| FFA-LoRA | 4 | 7 | 17 | 26 |

Table 25: The computational costs for each method on the Fed-Nature dataset for a single round of FL training.

| Method | # Per-round Computation Cost |
|---|---|
| Fedavg | 236s |
| Fedprox | 262s |
| FedAdam | 240s |
| FedAvgM | 241s |
| FedYogi | 239s |
| FFA-LoRA | 230s |

Table 26: Total transmitted message size (number of communication rounds × model parameters in millions) required by FedAvg to reach the target performance (accuracy 0.7 for Fed-SLAKE and 0.55 for Fed-FGVC) under varying straggler ratios, simulating heterogeneous environments. Results are reported for encoder-based VLMs with LLAMA3.2-3B as the baseline.

| Struggler% | Fed-SLAKE | Fed-FGVC |
|---|---|---|
| 0 | 24 ×23.9 | 23 ×23.9 |
| 10% | 25 ×23.9 | 25 ×23.9 |
| 30% | 26 ×23.9 | 27 ×23.9 |
| 50% | 28 ×23.9 | 34 ×23.9 |
| 70% | 27 ×23.9 | 38 ×23.9 |

