# OpenReview forum: "FedVLMBench: Benchmarking Federated Fine-Tuning of Vision-Language Models"
_ICLR.cc/2026/Conference — Submitted to ICLR 2026_

### Official Review · Reviewer_Jmzy · 2025-10-26

**Soundness:** 2
**Presentation:** 2
**Contribution:** 3
**Rating:** 2
**Confidence:** 4

**Summary:**

This paper presents FedVLMBench, the first systematic benchmark for federated fine-tuning of Vision-Language Models (VLMs), integrating two architectures (encoder-based and encoder-free), four fine-tuning strategies, six FL algorithms, and six datasets across single-task (VQA, captioning, classification, detection) and multi-task settings in natural and medical domains.

**Strengths:**

Innovation and Timeliness: FedVLMBench fills a gap in FL-VLMs benchmarks with novel multi-task datasets (Fed-Nature, Fed-Med) and expands task coverage from 2 to 4 types, providing timely value for advanced community research.

Theoretical and Practical Guidance: Findings highlight FL’s sensitivity to heterogeneity in vision-centric tasks and multi-task FL’s ability to mitigate non-IID issues, offering actionable research directions.

**Weaknesses:**

Motivation Insufficient: The Introduction notes that VLMs’ centralized training fails to meet privacy needs in healthcare and finance, yet lacks experimental validation of FL’s privacy capabilities, resembling a domain migration rather than a problem-driven contribution.

Gaps in Prior Work Lacking Depth: Motivation hinges on limited FL understanding across VLMs and a narrow task focus, essentially an extension of prior work; adding empirical evidence (e.g., why these gaps cause practical issues) would strengthen it.

Dataset Construction Limitations: Some datasets have small client numbers (e.g., 3 for Fed-SLAKE), deviating from typical FL scenarios (10-20+ clients), reducing usability and generalizability.

Limited Model Selection: Experiments rely on small models like LLAMA3.2-3B, lacking support for larger or diverse open-source models, limiting robustness. VLMs' performance is highly scale-dependent; this configuration underestimates real-world performance.

Resembles a Technical Report: The paper focuses on descriptive benchmarks and results but lacks deep theoretical discussion (e.g., why vision tasks are heterogeneity-sensitive), diminishing its research value.

**Questions:**

Please see the weaknesses above.

---

> ### Author Response · Authors · 2025-11-25
> **Response to Reviewer Jmzy**
>
> We thank the reviewer for acknowledging FedVLMBench's "Innovation" in bridging the FL-VLM benchmark gap through novel multi-task datasets and extended task coverage, as well as its practical value in delivering "actionable research directions". Please see below for our point-by-point responses to the specific comments.
>
>
> **Q1. Motivation Insufficient: The Introduction notes that VLMs’ centralized training fails to meet privacy needs, yet lacks experimental validation of FL’s privacy capabilities, resembling a domain migration rather than a problem-driven contribution.**
>
> We sincerely thank the reviewer for the thoughtful comments and appreciate the opportunity to clarify our motivation. First, we wish to clarify a potential misunderstanding: our primary motivation is not to simply combine FL and VLMs, nor to propose a new privacy-preserving algorithm.
>
> It is widely acknowledged that FL offers inherent privacy advantages due to its decentralized training paradigm, where only model weights—not raw data—are shared among participants [R1-R3]. As such, our work does not **seek to re-validate FL’s privacy capabilities experimentally, but rather to advance the field by providing the essential infrastructure for studying FL in the context of VLMs**. Specifically, our work is driven by the urgent need for a systematic and comprehensive benchmark for federated fine-tuning of VLMs, an area that is rapidly gaining importance but remains critically under-explored. **Below, we further summarize our motivation and the specific research gaps we address** (see the second paragraphs of our Introduction for more details).
>
> **(1)** existing VLMs can be categorized into two main types, encoder-based VLMs and encoder-free VLMs. These paradigms differ fundamentally in their components and fine-tuning mechanics. Current research in FL primarily focuses on designing client-specific adapters or new aggregation methods based on encoder-based VLM, but largely overlooks the emerging encoder-free paradigm and the extensive spectrum of fine-tuning strategies relevant to each architecture. This narrow perspective fails to address how differences in VLM architecture and fine-tuning strategies interact with data heterogeneity in FL, leaving a critical gap in understanding how to effectively apply FL across varied VLMs.
>
> **(2)** existing FL multimodal benchmark research focuses narrowly on two basic task types (VQA and classification) while ignoring more complex but critically important multimodal tasks such as report generation and visual localization.
>
> **(3)** no existing FL datasets support federated multi-modal multi-task learning scenarios, despite their practical significance in real-world applications where different clients may need to handle distinct multimodal tasks.
>
> In summary, our motivation is not a mere domain migration but rather the establishment of FedVLMBench, the first comprehensive benchmark suite for federated fine-tuning of VLMs. This work lays the foundational infrastructure for the community, directly enabling and accelerating innovation—much as previous benchmarks have done in other domains. We hope this clarifies our motivation and the significance of our work.
>
> [R1]McMahan B, et al. Communication-efficient learning of deep networks from decentralized data. PMLR, 2017.
>
> [R2]Zhang C, et al. A survey on federated learning[J]. Knowledge-Based Systems, 2021.
>
> [R3]Zhang K, et al. Challenges and future directions of secure federated learning: a survey. Frontiers of computer science, 2022.

---

> ### Author Response · Authors · 2025-11-25
> **Response to Reviewer Jmzy**
>
> **Q2: Motivation hinges on limited FL understanding across VLMs and a narrow task focus, essentially an extension of prior work; adding empirical evidence (e.g., why these gaps cause practical issues) would strengthen it.**
>
> We thank the reviewer for this valuable feedback. We respectfully argue that our work represents more than a simple extension of prior research, but rather addresses fundamental gaps in FL+VLM field through systematic investigation of multiple critical dimensions. Below, we clarify the existing challenges, our solutions, and the consequences if these gaps remain unaddressed.
>
> **(1) Diverse and Realistic Federated VLM Dataset Suite**
>
> **Challenge**: Prior FL multimodal benchmarks focus narrowly on basic tasks (VQA and classification), ignoring practically critical multimodal tasks like report generation and visual localization. Moreover, no existing datasets support federated multi-modal, multi-task learning, which is increasingly important in real-world deployments.
>
> **Our contribution**: We construct four cross-domain single-task datasets with configurable IID, simulated non-IID, and real non-IID splits, as well as two pioneering cross-domain, multi-task federated datasets—the first to enable collaborative tuning across interconnected multimodal objectives.
>
> **Impact**: Without these datasets, researchers cannot robustly evaluate methods under realistic conditions or assess robustness to heterogeneity, and would need to invest time building new datasets. Our suite provides immediate, standardized resources for comprehensive evaluation.
>
> **(2) A Unified and Public Codebase for FL-VLM Research**
>
> **Challenge**: Existing FL–VLM studies lack a unified, open-source codebase capable of supporting multiple VLM architectures, finetuning strategies, diverse FL algorithms, and heterogeneous cross-domain tasks. This forces researchers to re-implement and adapt tools, slowing progress and undermining fair comparison.
>
> **Our contribution**: We release FedVLMBench, an open-source platform dedicated to FL-VLM research, supporting 4 major VLM architectures, 7 FL algorithms, 6 cross-domain datasets (IID and non-IID), 4 fine-tuning strategies, and 4 downstream task categories.
>
> **Impact**: FedVLMBench will serve as a vital testbed for fair and extensible evaluation of future FL-VLM methods, accelerating future research.
>
> **(3) Comprehensive and Actionable In-depth Empirical Analysis**
>
> **Challenge**: Prior work offers a limited understanding of the interplay between VLM architectures, FL algorithms, tuning strategies, and heterogeneity in federated settings, especially for multi-task scenarios. This leaves practitioners uncertain about optimal strategies and unaware of key failure modes.
>
> **Our contribution**: Leveraging FedVLMBench, we provide the first in-depth empirical analysis of these interactions, yielding actionable guidelines (e.g., prioritize LLM tuning for text-centric tasks; connector tuning for vision-centric tasks; use a 2-layer MLP connector with concurrent connector-LLM tuning for encoder-based VLMs; multi-task FL serves as an alternative to centralized training while preserving privacy). We also identify critical challenges in FL-VLM, including the substantial sensitivity of current FL algorithms to non-IID vision-centric tasks across both encoder-based and encoder-free VLMs.
>
> **Impact**: Without these insights, practitioners may select suboptimal strategies and architectures, leading to inefficient training, degraded model performance, and resource waste in real-world deployments. Moreover, the community would remain unaware of key open challenges—such as the sharp performance degradation in vision-centric non-IID scenarios—thus hindering the development of more effective FL algorithms and overlooking promising directions like multi-task FL.
>
> We hope this response helps clarify your concern. We have also appropriately revised the manuscript to highlight the empirical evidence more explicitly.

---

> ### Author Response · Authors · 2025-11-25
> **Response to Reviewer Jmzy**
>
> **Q3. Some datasets have small client numbers (e.g., 3 for Fed-SLAKE), deviating from typical FL scenarios (10-20+ clients).**
>
> We thank the reviewer for the thoughtful feedback. We emphasize that our proposed datasets are carefully designed to cover a comprehensive spectrum of FL scenarios, with deliberate variations in both client scale (ranging from 3 to 30 clients) and task types (covering 4 single-task and 2 multi-task datasets spanning medical and natural domains). The numbers shown in Tab.2 refer to the **maximum number of real clients**, as partitioned according to actual data sources or tasks (as detailed in Section 3 and C.3 DATASETS in Appendix). For example, Fed-Med and Fed-Nature are split based on distinct real-world tasks, with each task assigned to a client. These real-world FL splits enable researchers to thoroughly investigate the performance of the model in diverse real-world data distributions. More importantly, to fully support the study of algorithms in larger-scale settings, our released codebase provides a flexible FL client simulation method based on the widely used Dirichlet distribution strategies. Researchers can easily use our provided scripts to simulate a larger number of clients (e.g., 20, 50, 100, or more) from all datasets. We have now revised the manuscript to better reflect this design.

---

> ### Author Response · Authors · 2025-11-25
> **Response to Reviewer Jmzy**
>
> **Q4. Limited model selection without larger or diverse open-source models.**
>
> We thank the reviewer for raising this important point regarding model selection. In our original manuscript, we evaluate three VLMs: two encoder-based models (**LLAMA3.2-3B and Qwen2.5-VL-3B**) and one encoder-free model (**Show-O-3B**). As noted in Lines 315-318, we further assess the generalization of our key findings on Qwen2.5-VL-3B, with results detailed in Tab.10 and 11 (Appendix D.4.1).
>
> To further address this concern and strengthen our findings, we conduct additional experiments with two more open-source VLMs: **Qwen2.5-VL-7B** (encoder-based, large scale) and **Mono-InternVL-3B**[R4] (encoder-free). We focus on validating our main takeaways 5 and 6 across three representative datasets: the text-centric Fed-SLAKE, the vision-centric Fed-FGVC, and the multi-task Fed-Nature.
>
> As illustrated in Tab.A17, we evaluate the performance on Fed-SLAKE and Fed-FGVC datasets under both IID and non-IID settings. For the text-centric task (Fed-SLAKE), both FL methods perform similarly, regardless of the data distribution (IID or non-IID). In contrast, the vision-centric task reveals a clear performance decline under non-IID conditions. This finding validates the conclusion presented in Takeaway 5. On the multi-task Fed-Nature dataset (Tab.A18), federated multi-modal learning achieves results comparable to centralized training, particularly for FedAvg, in line with Takeaway 6. We also repeat the same experiments with the larger Qwen2.5-VL-7B model. Importantly, our experiments with this larger Qwen2.5-VL-7B model (see Tab.A19 and Tab.A20) further confirm that both Takeaway 5 and Takeaway 6 generalize well to large-scale encoder-based VLMs.
>
> In summary, our extended experiments across multiple model scales and architectures—including encoder-based VLMs (**LLAMA3.2-3B, Qwen2.5-VL-3B, Qwen2.5-VL-7B**) and encoder-free VLMs (**Show-O-3B and Mono-InternVL-3B**)—consistently validate our core findings. These results strengthen the general applicability of the takeaways proposed in our study. We have incorporated these experimental results into our manuscript.
>
>
> Table A17: Performance comparison of Mono-InternVL on two representative single-task datasets with IID and non-IID distributions.
> |  Method  | Fed-SLAKE | Fed-SLAKE | Fed-FGVC | Fed-FGVC |
> |:-|:-:|:-:|:-:|:-:|
> |  |  IID   | Non-IID |   IID  | Non-IID |
> | Central  |  0.673 |    —    |  0.561 |    —    |
> | FedAvg   |  0.621 |   0.615 |  0.504 |  0.401  |
> | FFA-LoRA |  0.622 |   0.616 |  0.514 |  0.402  |
>
> Table A18:Performance comparison of Mono-InternVL on Fed-Nature dataset. MT-Central refers to centralized training on the centralized multi-task dataset.
> | Mode | VQA | Caption Generation | Visual Grounding | Classification |
> |:-|:-:|:-:|:-:|:-:|
> | MT-Central| 0.686 | 1.020/0.398 | 0.326  | 0.911 |
> | FedAvg    | 0.711 | 1.080/0.406 | 0.337  | 0.930 |
> | FFA-LoRA  | 0.663 | 1.037/0.398 | 0.312  | 0.906 |
>
>
> Table A19: Performance comparison of Qwen2.5-VL-7B on two representative single-task datasets with IID and non-IID distributions.
> | Method  | Fed-SLAKE | Fed-SLAKE | Fed-FGVC | Fed-FGVC |
> |:-|:-:|:-:|:-:|:-:|
> | |  IID  |  Non-IID  |   IID  | Non-IID |
> | Central  |  0.808  |      — |   0.913  |      — |
> | FedAvg   |  0.781  |  0.785   |   0.885 |  0.851  |
> | FedProx  |  0.768  |  0.766   |   0.880 |  0.843  |
> | FFA-LoRA |  0.746  |  0.750   |   0.869  |  0.796  |
>
> Table A20:Performance comparison of Qwen2.5-VL-7B on Fed-Nature dataset. MT-Central refers to centralized training on the centralized multi-task dataset.
> | Mode | VQA | Caption Generation | Visual Grounding | Classification |
> |:-|:-:|:-:|:-:|:-:|
> | MT-Central| 0.823 | 1.545/0.463 | 0.675 | 0.982 |
> | FedAvg    | 0.820 | 1.542/0.462 | 0.647 | 0.981 |
> | FFA-LoRA  | 0.791 | 1.537/0.457 | 0.598 | 0.976 |
>
> [R4]Luo G, et al. Mono-internvl: Pushing the boundaries of monolithic multimodal large language models with endogenous visual pre-training. CVPR. 2025.

---

> ### Author Response · Authors · 2025-11-25
> **Response to Reviewer Jmzy**
>
> **Q5. Resembles a technical report without deep theoretical discussion (e.g., why vision tasks are heterogeneity-sensitive).**
>
> We thank the reviewer for this valuable feedback. We respectfully argue that the primary contribution of our work is not to propose a new algorithm with theoretical analysis, but rather to establish FedVLMBench as the first comprehensive benchmark specifically tailored for the emerging field of federated fine-tuning of VLMs. In rapidly evolving fields, such rigorous and standardized benchmarks are essential for enabling and accelerating technical innovation, as evidenced by the significant impact of similar benchmark papers at top venues.
>
> Our FedVLMBench offers three **key contributions** to the FL community: **1) a novel and extensive federated VLM dataset suite, 2) a unified and public codebase for FL-VLM research, and 3) comprehensive and in-depth empirical analyses with new findings**, offering actionable guidelines for practitioners and revealing open challenges for the community.  All of these contributions are, to our knowledge, new to the FL community and are also firmly supported by other reviewers. For example, Reviewer J2Ci recognizes the value of "providing empirical guidelines for federated VLM deployment in privacy-sensitive domains," Reviewer yqY5 notes the "valuable insights and contributions to the research community," and Reviewer egwi affirms that “both benchmark and findings are valuable for researchers in the community”. Please refer to our responses to Q1 and Q2 for a detailed discussion of our motivation and the significance of our contributions.
>
> In response to the reviewer’s concern regarding the lack of deep theoretical discussion, particularly about why vision-centric  tasks are more sensitive to data heterogeneity than text-centric tasks, we have now expanded our key findings to clarify the underlying mechanisms.  We hypothesize that this is because text-centric tasks can leverage the strong language priors embedded in pre-trained LLMs, making them more robust to heterogeneous data. To validate this, we conduct two key analyses:
>
> **(1) The Impact of Visual Inputs on Performance in Text-centric Task and Vision-centric Task**: We first evaluate the performance of both vision- (Fed-FGVC)  and text-centric tasks (Fed-SLAKE) under varying degrees of image occlusion (0%, 20%, 50%, 80%, and 99%). As shown in Tab.A21, the vision-centric task suffers a sharp performance drop (−84.5%) even under mild occlusion (20%), while the text-centric task declines by only 38.2% even under near-total occlusion (99%). This strongly suggests that text-driven tasks can rely on pre-trained knowledge to compensate for missing visual information, thereby reducing their sensitivity to data heterogeneity.
>
> **(2) The Impact of Pre-trained LLMs in Mitigating Data Heterogeneity**: To confirm whether pre-trained language priors are responsible for the robustness of text-centric tasks, we compare the performance of text-centric tasks under both IID and non-IID settings, where the LLM is either initialized from pre-trained weights or from random initialization (see Tab.A22). As expected, the randomly initialized models suffer much greater performance degradation under non-IID data (7.19% degradation), highlighting the importance of pre-trained linguistic knowledge in mitigating the negative effects of data heterogeneity.
>
> We thank the reviewer again for raising this insightful point. We have included these deep analyses in the revised manuscript to further clarify and strengthen our empirical findings.
>
> Table A21: The impact of different image occlusion rates on Show-O's Performance in the text-centric task (Fed-SLAKE dataset) and the vision-centric task (Fed-FGVC dataset).
> | Occlusion rate | Fed-SLAKE | Fed-FGVC |
> |:-|:-:|:-:|
> | 0    | 0.784         | 0.739         |
> |20%|0.657(-16.2%)|0.114(-84.5%)|
> |50%|0.545(-30.4%)|0.049(-93.3%)|
> |80%|0.509(-35.1%)|0.036(-95.1%)|
> |99%|0.484(-38.2%)|0.019(-97.4%)|
>
> Table A22:  Effect of LLM pre-training versus random initialization on text-centric tasks (Fed-SLAKE dataset).
> | Method | Fed-SLAKE | Fed-SLAKE |
> |:-|:-:|:-:|
> |                |  IID  | Non-IID |
> | Pre-trained Show-O|0.777|0.761|
> | Random init Show-O   | 0.681 |  0.632  |

---

### Official Review · Reviewer_egwi · 2025-10-26

**Soundness:** 3
**Presentation:** 3
**Contribution:** 2
**Rating:** 6
**Confidence:** 3

**Summary:**

Summary:

This paper aims to provide a benchmark for federally finetuning VLM, which covers 2 architectures (encoder-based / encoder-free), 4 tuning strategies, 6 FL algorithms, and 6 datasets with different settings. The main findings from the paper are that: 1) for encoder-based VLM, a lightweight (typically 2-layer) MLP connector is the best; 2) joint tuning of connector and LLM works better than two-stage training. 3) Text-centric tasks benefit from LLM tuning while vision-centric tasks need connector tuning. 4) Non-IID hurts vision tasks much more. 5) Federated multi-task can reach near centralized level. Overall, both benchmark and findings are valuable for researchers in the community.

**Strengths:**

Strength:

1. The paper considered a wide range of downstream tasks, 2 main VLM architectures, 4 mainstream finetuning strategies, and multiple datasets across different domains, which makes the benchmark more comprehensive.
2. While current VLMs are strong on natural image and language tasks, they still underperform on domain-specific tasks like medical imaging. Adapting VLMs faces two real issues: limited data and strict privacy. This paper gives a practical, comprehensive guideline to personalize VLMs for the medical domain using federated fine-tuning.

**Weaknesses:**

Weakness & question:

1. While the considered VLM approaches are very up-to-dated, most compared baselines are from 5 years ago. Is there any reason for choosing these methods?
2. While the 5 takeaways seems very valuable and reasonable, I'm interested in takeaway 1 - why 2-layer should be better than 6-layer? Do you have any insight or analysis experiments to further explore this phenomenon?
3. In domains that have privacy concerns, like heathcare, the fairness problem is very pronounced.  While this paper focuses on benchmarking VLM finetuning methods, it's worth discussing the fairness issues and potential methods to address them in related works [1-4].
4. Another suggestion is that, non-iid is a general assumption in federated learning, so it would be better to show do takeaways 1,2,3,5 still hold under data heterogeneity. Given the computational limitation, it's fine to just provide some analysis and leave this for future work.
5. Although federated finetune VLM is a good research question in my opinion, adding results on decentralized and centralized training baselines in the main table would help readers to understand the importance of doing federated finetuning. [mainly in table 4 & 5, table 3]
6. It's also better to provide a figure to illustrate pipelines/workflows of each federated learning strategy; people in VLM or the medical domain may not be familiar with different FL strategies' settings.



[1] Huang W, Ye M, Shi Z, et al. Federated learning for generalization, robustness, fairness: A survey and benchmark[J]. IEEE Transactions on Pattern Analysis and Machine Intelligence, 2024, 46(12): 9387-9406.

[2] Zhang F, Shuai Z, Kuang K, et al. Unified fair federated learning for digital healthcare[J]. Patterns, 2024, 5(1).

[3] Wang H, Chen W, Luo X, et al. Toward Fair and Accurate Cross-Domain Medical Image Segmentation: A VLM-Driven Active Domain Adaptation Paradigm[C]//Proceedings of the IEEE/CVF International Conference on Computer Vision. 2025: 24102-24112.

[4] Zeng H, Yue Z, Zhang Y, et al. Fair federated learning with biased vision-language models[C]//Findings of the Association for Computational Linguistics ACL 2024. 2024: 10002-10017.

**Questions:**

see weakness

---

> ### Author Response · Authors · 2025-11-25
> **Response to Reviewer egwi**
>
> We sincerely thank the reviewer for your efforts and positive feedback. We are glad to be recognized for presenting a “comprehensive” benchmark with practical value, and for "a practical, comprehensive guideline to personalize VLMs for the medical domain". Please see our response below regarding the specific comments.
>
>
> **Q1. The considered VLM approaches are very up-to-dated, most compared baselines are from 5 years ago. Is there any reason for choosing these methods?**
>
> We thank the reviewer for this careful observation and the valuable suggestion. The classical FL methods included in our study, such as FedAvg, FedAvgM, and FedProx, are widely adopted and continue to serve as standard baselines in recent literature. As suggested, in addition to the recent FFA-LoRA[R1] method designed for large model finetuning, we have now incorporated FedSA-LoRA[R2]—a latest FL method—into FedVLMBench, with comprehensive results provided in Tab.A11, A12, and A13. Importantly, the inclusion of this new method does not affect our main takeaways; all key findings remain consistent.
>
>
> Table A11: Performance comparison of different VLM architectures on four single-task datasets with IID and non-IID distributions.
> |  Mode | Method | Fed-SLAKE | Fed-SLAKE | Fed-ScienceCap|Fed-ScienceCap|Fed-FGVC | Fed-FGVC |Fed-RadGnome|Fed-RadGnome|
> |:-|:-|:-:|:-:|:-:|:-:|:-:|:-:|:-:|:-:|
> |  | |  IID  | Non-IID |   IID  | Non-IID |IID  | Non-IID |IID  | Non-IID |
> |Encoder-based|FedSA-LoRA  |  0.817  |  0.806  |  7.524/0.899  |  7.498/0.897  | 0.727  |  0.573  |  0.546  |  0.502  |
> |Encoder-free|FedSA-LoRA   |  0.640  |  0.631  |  6.005/0.837  | 5.956/0.819   | 0.477  |   0.304 |  0.515  |   0.483  |
>
>
> Table A12:Performance comparison of different VLM architectures on Fed-Nature dataset.
> |  Mode | Method | VQA | Caption Generation | Visual Grounding | Classification |
> |:-|:-|:-:|:-:|:-:|:-:|
> |Encoder-based| FedSA-LoRA |  0.749    | 0.806/0.349  | 0.371  |  0.924   |
> |Encoder-free| FedSA-LoRA  |  0.653    | 0.423/0.236  |  0.0   |  0.768   |
>
> Table A13:Performance comparison of different VLM architectures on Fed-Med dataset.
> |  Mode | Method| VQA | Report Generation | Detection |
> |:-|:-|:-:|:-:|:-:|
> |Encoder-based| FedSA-LoRA  |  0.707  |  1.819/0.571   |   0.572 |
> |Encoder-free | FedSA-LoRA  |  0.418  |  0.402/0.436   |   0.0   |
>
> Note: FedSA-LoRA is optimized for personalized FL scenarios, which explains its limited performance in some global-model-centric evaluations.
>
> [R1]Sun Y, et al. Improving lora in privacy-preserving federated learning. ICLR, 2024.
>
> [R2]Guo P, et al. Selective aggregation for low-rank adaptation in federated learning. ICLR, 2025.
>
>
> **Q2. Any insight or analysis experiments to further explore phenomenon: why 2-layer should be better than 6-layer?**
>
> We thank the reviewer for raising this insightful and important question. This observation is consistent with prior findings in centralized training settings [R3–R4], which have shown that simpler linear projectors or shallow MLPs often outperform deeper MLP- or Transformer-based connectors.
>
> The main reason, as we understand, lies in the role of the connector within encoder-based VLMs. Since both the visual encoder and the LLM are already powerful and well-pretrained, the connector's primary function is to *project* visual features into the LLM's semantic space, rather than perform complex vision feature extraction or transformation. Therefore, a 2-layer MLP provides sufficient non-linearity to align visual and textual spaces without introducing unnecessary complexity.
>
> In contrast, a 6-layer MLP introduces substantial parameter redundancy and increases model complexity. This additional depth may lead to over-processing of the visual features, resulting in task-specific representations that may deviate from the LLM’s semantic space, undermining the alignment objective and potentially harming generalization. Moreover, in the FL setting, where data distributions across clients may differ, the 6-layer MLP's increased capacity makes it more prone to memorizing client-specific noise and statistical biases. This effect can further widen the performance gap between the 2-layer and 6-layer MLPs in FL scenarios, as the deeper model is less robust to heterogeneity and more likely to overfit to local data. This is also evident in our results in Tab.3: while the 6-layer MLP can occasionally achieve slightly better performance than the 2-layer MLP in the centralized setting (e.g., F-2stage of Central on the Fed-ScienceCap dataset in Tab.3), the 2-layer MLP consistently outperforms the 6-layer MLP in FL settings.
>
> We hope this analysis provides a clearer understanding of the underlying phenomenon. We have incorporated this discussion into the revised manuscript to clarify our reasoning.
>
> [R3]Lin J, Yin H, Ping W, et al. Vila: On pre-training for visual language models. CVPR 2024.
>
> [R4]Li W, et al. Lost in embeddings: Information loss in vision-language models[J]. arXiv, 2025.

---

> ### Author Response · Authors · 2025-11-25
> **Response to Reviewer egwi**
>
> **Q3. The fairness problem is very pronounced, it's worth discussing the fairness issues and potential methods to address them in related works [1-4].**
>
> We thank the reviewer for highlighting the importance of fairness in FL. We have now incorporated a dedicated discussion of fairness issues and future directions in the revised "Limitation & Further Work" section of our manuscript. See below for the addition in Discussion:
>
> Furthermore, fairness remains an important yet underexplored issue in our current benchmark. While we have focused on overall performance metrics, we acknowledge that FL applications in privacy-sensitive domains such as healthcare often face pronounced fairness challenges, such as performance disparities across demographic groups and institutions. The integration of VLMs into federated settings may further amplify such issues due to biases inherited from large-scale pretraining and data distribution shifts among clients. Recent studies [R5-R8] have discussed and proposed fairness-enhancing strategies for FL, including fairness-aware optimization and bias mitigation techniques. These efforts generally follow two technical pathways: evaluation-and-optimization based mechanisms for system-level fairness, and data/feature-level interventions using VLM-specific methods like attribute-aware sampling and visual prompt tuning. However, systematic investigation of fairness in FL-VLM settings, and the development of dedicated methods for mitigating bias in federated finetuning of VLMs, remain largely open problems. We hope our benchmark can serve as a foundation for advancing research in fair and equitable FL-VLM systems.
>
> [R5] Huang W, Ye M, Shi Z, et al. Federated learning for generalization, robustness, fairness: A survey and benchmark. TPAMI 2024.
>
> [R6] Zhang F, Shuai Z, Kuang K, et al. Unified fair federated learning for digital healthcare. Patterns, 2024.
>
> [R7] Wang H, Chen W, Luo X, et al. Toward Fair and Accurate Cross-Domain Medical Image Segmentation: A VLM-Driven Active Domain Adaptation Paradigm. ICCV 2025.
>
> [R8] Zeng H, Yue Z, Zhang Y, et al. Fair federated learning with biased vision-language models. ACL 2024.
>
> **Q4. Non-iid is a general assumption in federated learning, so it would be better to show do takeaways 1,2,3,5 still hold under data heterogeneity.**
>
> We thank the reviewer for this valuable suggestion. We are pleased to clarify that most of our takeaways (specifically Takeaways 2, 3, and 5) have been validated through experiments conducted under non-IID data settings, as shown in Tab.4 and Tab.5. For Takeaway 1, we now supplement experiments following the same non-IID configuration as Tab.3. As shown in the newly added Tab.A14, the MLP2x connector remains the most efficient choice under data heterogeneity.
>
> Table A14: Performance comparison of connector layer types (linear layer, 2-layer MLP (Mlp2x), and 6-layer MLP (Mlp6x)) on FL fine-tuning on encoder-based VLM undering Non-IID data portions of Fed-SLAKE and Fed-ScienceCap datasets.
> |  Mode | Method| Fed-SLAKE | Fed-SLAKE | Fed-SLAKE|Fed-ScienceCap|Fed-ScienceCap | Fed-ScienceCap |
> |:-|:-|:-:|:-:|:-:|:-:|:-:|:-:|
> | | |  Linear| Mlp2x |   Mlp6x  | Linear  | Mlp2x |   Mlp6x|
> |F-C|Fedavg|0.712|0.775|0.744|7.037/0.863|7.249/0.881|6.998/0.866|
> |F-L|Fedavg|0.775|0.802|0.785|7.338/0.888|7.342/0.889|7.182/0.885|
> |F-CL|Fedavg|0.821|0.827| 0.805|7.404/0.893|7.476/0.897|7.150/0.882|
> |F-2stage|Fedavg|0.815|0.814| 0.810|7.140/0.874|7.281/0.883|7.258/0.880|

---

> > ### Comment · Reviewer_egwi · 2025-11-26
> >
> > Thank you for your reply! I don't have any other technical concerns, so I will raise the confidence score to 4.

---

> > > ### Author Response · Authors · 2025-11-26
> > > **Response to Reviewer egwi**
> > >
> > > Thank you for your response and for taking the time to thoroughly review our work! We are pleased that we have addressed your technical concern, and we greatly appreciate your ongoing support, as your suggestions are invaluable to us.
> > >
> > > If there are any additional questions or remaining issues, we would be more than happy to provide further clarifications.  We sincerely hope that, with your concerns addressed and the improvements made, our work could merit your increased support.
> > >
> > > Once again, we deeply appreciate your time and constructive feedback, which has significantly improved our paper.

---

> ### Author Response · Authors · 2025-11-25
> **Response to Reviewer egwi**
>
> **Q5. Adding results on centralized training baselines in the main table would help readers to understand the importance of doing federated finetuning. [mainly in table 4 & 5, table 3]**
>
> We sincerely thank the reviewer for this insightful suggestion. We provide centralized training result in Tab.A15 and Tab.A16, and update Tab.4 and Tab.5 with these results (while Tab.3 and Tab.6 already included such comparisons). After including these baselines, we found that our optimal selection—F-CL on encoder-based VLMs and federated finetuning of VLMs under IID settings—generally achieves results comparable to the centralized baseline (ceiling) performance. This further demonstrates the effectiveness of federated finetuning for VLMs. We greatly appreciate the reviewer’s insightful comments, which help us better demonstrate the value of federated finetuning of VLMs.
>
> Table A15: Quantitative comparison of four fine-tuning strategies on multi-type task datasets with central training.
> |  Mode | Method| Fed-SLAKE | Fed-ScienceCap | Fed-FGVC|
> |:-|:-|:-:|:-:|:-:|
> |F-C|Central|0.788|7.361/0.889|0.751|
> |F-L|Central|0.834|7.459/0.896|0.707|
> |F-CL|Central|0.839|7.550/0.901| 0.764|
> |F-2stage|Central|0.845|7.591/0.908| 0.772|
>
> Table A16: Performance comparison of different VLM architectures on various single-task datasets with
> central training.
> |Mode| Method| Fed-SLAKE | Fed-ScienceCap | Fed-FGVC| Fed-RadGnome|
> |:-|:-|:-:|:-:|:-:|:-:|
> |Encoder-based|Central|0.839|7.550/0.901| 0.764|0.594|
> |Encoder-free |Central|0.784|7.462/0.899|0.739|0.612|
>
> **Q6. It's also better to provide a figure to illustrate pipelines/workflows of each federated learning strategy; people in VLM or the medical domain may not be familiar with different FL strategies' settings.**
>
> We thank the reviewer for this valuable suggestion. We agree that a clear visual representation of the FL methods will greatly enhance the accessibility of our work for audiences from both the VLM and medical domains. In response, **we have provided a new figure to the revised Appendix C.4 (see Figure 4 in the Appendix), which illustrates the workflows of the key FL strategies discussed in the main paper.** Specifically, the upper part of the figure provides a step-by-step overview of the basic FL process, making it easier for readers to understand how FL operates in practice. It also highlights two main approaches for addressing data heterogeneity: (A) regularizing local training (e.g., FedProx) and (B) improving global aggregation strategies (e.g., FedAdam, FedYogi, FedAvgM). The lower part of the figure visually compares different parameter-sharing strategies in LoRA-based FL, including LoRA, FFA-LoRA, and FedSA-LoRA, clarifying which components are trainable and shared in each case. We believe this detailed illustration will help readers more clearly understand both the general FL workflow and the differences among the FL methods studied in our work.

---

### Official Review · Reviewer_yqY5 · 2025-10-28

**Soundness:** 2
**Presentation:** 3
**Contribution:** 2
**Rating:** 4
**Confidence:** 5

**Summary:**

This paper focuses on benchmarking the integration of Vision-Language Models (VLMs) with Federated Learning (FL) and identifies several key limitations in existing benchmarks: the limited range of VLM architectures studied, the restricted task diversity, and the lack of systematic investigation into the relationships among connectors, LoRA, and data heterogeneity. To address these issues, the paper explores two types of VLM architectures—encoder-based and encoder-free, and extends the benchmark to include a wider variety of downstream tasks, such as report generation. It further examines different fine-tuning combinations of connectors and LoRA, revealing several important experimental phenomena and insights.

**Strengths:**

This benchmark is meaningful as it incorporates a wider variety of VLM architectures, a more diverse set of tasks, and different fine-tuning strategies for VLMs under the Federated Learning (FL) framework. Such a design enhances the comprehensiveness and practical value of the evaluation, providing valuable insights and contributions to the research community.

**Weaknesses:**

Although the benchmark is meaningful, it still has several shortcomings:

1. Experimental aspect: It only presents the performance of different VLMs across various tasks and fine-tuning strategies. However, the experimental results, especially under different modes (F-C, F-L, F-CL, F-2stage), show limited differences, and it is unclear whether testing such combinations of fine-tuning strategies is truly necessary for FL;
2. Technical aspect: The work lacks technical innovation. It merely evaluates more tasks and provides conclusions without deeply analyzing the issues that arise in different tasks or proposing any technical solutions to address them;
3. Model aspect: Although it claims to cover both encoder-based and encoder-free VLMs, only two specific models are actually tested, which is insufficient to represent the diversity within these two categories;
4. Federated Learning aspect: Key factors that the FL community cares about, such as heterogeneous environments and varying numbers of clients, are not well explored or reflected in the benchmark.

**Questions:**

The main issues lie in the fact that this benchmark does not thoroughly identify the limitations across different tasks, VLM types, and fine-tuning strategies, nor does it propose technically innovative solutions to address these shortcomings. Moreover, the performance differences among various fine-tuning strategies are minimal, making it difficult to fully support the conclusions drawn. In addition, the benchmark fails to address the core concerns of the Federated Learning community, and the range of model architectures included remains quite limited.

---

> ### Author Response · Authors · 2025-11-25
> **Response to Reviewer yqY5**
>
> We thank the reviewer for the time and effort in reviewing our paper and providing constructive comments. We are pleased that the reviewer recognized the “meaningful” contribution of our benchmark, particularly its “comprehensive evaluation”. Please see our response below regarding the specific comments.
>
> **Q1. The experimental results, especially under different modes (F-C, F-L, F-CL, F-2stage), show limited differences。 It is unclear whether testing such combinations of fine-tuning strategies is truly necessary for FL**
>
> We sincerely thank the reviewer for their careful observation and insightful comment. We agree that from a pure accuracy perspective, the differences between some fine-tuning strategies might appear limited at first glance. However, in the context of FL, the evaluation must extend beyond accuracy to include **communication efficiency**—the primary bottleneck in FL systems. Our systematic comparison reveals several non-trivial and actionable findings that are critical for practical FL deployment:
>
> 1) The more complex two-stage finetuning strategy (F-2stage) often does not yield consistent performance gains over the simpler joint tuning of the connector and LLM (F-CL) (limited accuracy differences, see Tab.4 in manuscript). Yet, F-2stage incurs significantly **higher communication overhead**, as shown in Tab.A3. This makes the simpler F-CL a strictly superior choice from a practical systems perspective.
>
> 2) Furthermore, the choice between tuning the connector (F-C) or the LLM (F-L) shows **substantial and consistent performance differences** tied to task nature (see Tab.4 in manuscript). For text-centric tasks (e.g., VQA on Fed-SLAKE), tuning the LLM (F-L) is markedly more effective (e.g., 0.806/0.802 compared to 0.783/0.775 of FedAvg on Fed-SLAKE).  In contrast, for vision-centric tasks, tuning the connector is clearly preferable (e.g., 0.724/0.585 vs. 0.647/0.529 of FedAvg on Fed-FGVC).
>
> Therefore, systematically testing these fine-tuning combinations allows us to provide clear and actionable guidelines: practitioners can confidently adopt the efficient F-CL as a default strategy and further specialize by selecting the tuning component based on task type—F-L for text-centric tasks and F-C for vision-centric ones. This approach directly aids the FL community in making resource-efficient decisions without sacrificing model performance.
>
> Thank you again for this thoughtful feedback. We have revised the manuscript to better highlight the practical value of our findings.
>
> Table A3: Total transmitted message size (number of communication rounds and model parameters transmission (GB) in all) required by FedAvg to reach the target performance (accuracy 0.8 for Fed-SLAKE and 0.7 for Fed-FGVC) based on encoder-based VLM (LLaVA-1.5-3B)
> |Mode|Method| Fed-SLAKE(0.8)  | Fed-FGVC(0.7) |
> |:-|:-:|:-:|:-:|
> | F-CL     | FedAvg |  15 (6.69)  | 33 (14.72) |
> | F-2stage | FedAvg |  41 (9.31)  | 72 (16.11) |

---

> ### Author Response · Authors · 2025-11-25
> **Response to Reviewer yqY5**
>
> **Q2. Insufficient experimental comparisons and no technically innovative solutions. It only presents the performance without deep analysis or proposing any technical solutions.**
>
> We sincerely thank the reviewer for this valuable feedback and appreciate the opportunity to further clarify the scope, depth, and significance of our work.
>
> **1. Unique Value of FedVLMBench**
>
> Our primary goal is not to propose a new algorithm, but to establish FedVLMBench as the first comprehensive benchmark tailored for the emerging field of FL-VLM. Our core contributions are:
>
> (1) Novel FL-VLM datasets with 4 cross-domain single-task datasets and 2 pioneering cross-domain multi-task federated datasets. To our knowledge, these are the first to enable federated instruction tuning across multimodal objectives, addressing a crucial void for real-world FL-VLM deployment.
>
> (2) An open-source codebase dedicated to FL-VLM research supporting multiple VLM architectures, 7 FL algorithms, 6 cross-domain datasets spanning IID and non-IID data distributions, 4 fine-tuning strategies, and 4 distinct downstream task categories, providing a fair and extensible testbed for future FL-VLM research.
>
> (3) Comprehensive empirical analysis: Our work **moves far beyond superficial model performance reporting**. Instead, we systematically analyze the interaction between VLM architectures, FL algorithms, fine-tuning strategies, heterogeneity, and multi-task learning. Our findings, rigorously validated through extensive generalization studies (e.g., varying VLM types and LoRA ranks), yield **novel and actionable insights for the FL community**. For instance, we reveal that current FL methods exhibit higher sensitivity to data heterogeneity in vision-centric tasks than text-centric ones. To our knowledge, all these established findings are new to the FL field. It not only provides actionable guidelines for practitioners, but also reveals open challenges for the community, **directly inspiring future algorithmic innovation.**
>
>
> **2. Strengthening the Benchmark with In-Depth Analysis and New Experiments**
>
> We fully agree that a thorough analysis and broad empirical evaluation are essential for a high-quality benchmark. Therefore, we have substantially expanded both the depth and breadth of our study:
>
> (1) **In-depth Analysis of Key Findings**
>
> We elaborate on why text-centric tasks are less sensitive to data heterogeneity than vision-centric tasks. In our main paper, we hypothesize that this is because text-driven tasks inherently benefit from the strong language priors of pre-trained LLMs. To validate this, we conduct two analyses:
>
> a) Sensitivity to Visual Occlusion: We first evaluate the performance of vision (Fed-FGVC) and text-centric tasks (Fed-SLAKE) under varying image occlusion rates (0%, 20%,..., 99%). As shown in Tab.A4, the vision-centric task suffers a sharper performance drop than the text-centric task. This suggests that text-driven tasks can leverage pre-trained knowledge to compensate for missing visual information, thereby reducing their sensitivity to data heterogeneity.
>
> b) Role of Pre-trained LLMs: To confirm whether pre-trained language priors are responsible for the robustness of text-centric tasks, we compare the performance of text-centric tasks under both IID and non-IID settings using either pre-trained or randomly initialized LLMs in Tab.A5. As expected, models trained from random initialization suffer much greater performance degradation under non-IID data (7.19% drop), highlighting the importance of pre-trained linguistic knowledge in mitigating the negative effects of data heterogeneity.
>
> Table A4: Impact of image occlusion rate on Show-O model performance with FedAVG in text-centric (Fed-SLAKE dataset) and vision-centric (Fed-FGVC dataset) tasks.
> |Occlusion rate|Fed-SLAKE|Fed-FGVC|
> |:-|:-:|:-:|
> |0|0.784|0.739|
> |20%|0.657(-16.2%)|0.114(-84.5%)|
> |50%|0.545(-30.4%)|0.049(-93.3%)|
> |80%|0.509(-35.1%)|0.036(-95.1%)|
> |99%|0.484(-38.2%)|0.019(-97.4%)|
>
> Table A5: Impact of LLM pre-training on text-centric task (Fed-SLAKE dataset).
> |Model|Fed-SLAKE|Fed-SLAKE|
> |:-|:-:|:-:|
> ||IID|Non-IID|
> |Pre-trained Show-O|0.777|0.761|
> |Random Init Show-O|0.681|0.632|
>
> (2) **Expanded Experiments**
>
> To address concerns about experimental breadth, we expand FedVLMBench by (i) supporting more VLM architectures, (ii) evaluating under realistic system heterogeneity, and (iii) including a recent FL algorithm (FedSA-LoRA, ICLR 2025). Please refer to our responses to Q3 (new VLM architectures), Q4 (system heterogeneity), and Q1 of Reviewer egwi (new FL algorithm) for details.
>
> In summary, our FedVLMBench delivers new datasets, code infrastructure, novel empirical insights, and practical guidelines that collectively form an essential resource for the FL community. We believe that high-quality benchmarks and rigorous empirical analysis—especially at this scale—are as vital as new theoretical algorithms. We hope this clarifies the reviewer's concerns.

---

> ### Author Response · Authors · 2025-11-25
> **Response to Reviewer yqY5**
>
> **Q3. Only two specific models are actually tested, which is insufficient to represent the diversity within these the encoder-based and encoder-free VLM categories;**
>
> We appreciate the reviewer’s concern regarding model diversity. In our original manuscript, we evaluat three VLMs: two encoder-based models (**LLAMA3.2-3B and Qwen2.5-VL-3B**) and one encoder-free model (**Show-O-3B**). As noted in Lines 315-318, we further assess the generalization of our key findings on Qwen2.5-VL-3B, with results detailed in Tables 10 and 11 (Appendix D.4.1).
>
> To further address this concern and strengthen our findings, we conduct additional experiments with two more open-source VLMs: **Qwen2.5-VL-7B** (encoder-based, large scale) and **Mono-InternVL-3B**[R1] (encoder-free). We focus on validating our main takeaways 5 and 6 across three representative datasets: the text-centric Fed-SLAKE, the vision-centric Fed-FGVC, and the multi-task Fed-Nature.
>
> As illustrated in Tab.A6, we evaluate the performance on Fed-SLAKE and Fed-FGVC datasets under both IID and non-IID settings. For the text-centric task (Fed-SLAKE), both FL methods perform similarly, regardless of the data distribution (IID or non-IID). In contrast, the vision-centric task reveals a clear performance decline under non-IID conditions. This finding validates the conclusion presented in Takeaway 5. On the multi-task Fed-Nature dataset (Tab.A7), federated multi-modal learning achieves results comparable to centralized training, particularly for FedAvg, in line with Takeaway 6. We also repeat the same experiments with the larger Qwen2.5-VL-7B model. Importantly, our experiments with this larger Qwen2.5-VL-7B model (see Tab.A8 and Tab.A9) further confirm that both Takeaway 5 and Takeaway 6 generalize well to large-scale encoder-based VLMs.
>
> In summary, our extended experiments across multiple model scales and architectures—including encoder-based VLMs (**LLAMA3.2-3B, Qwen2.5-VL-3B, Qwen2.5-VL-7B**) and encoder-free VLMs (**Show-O-3B and Mono-InternVL-3B**)—consistently validate our core findings.
>
>
> Table A6: Performance comparison of Mono-InternVL-3B on two representative single-task datasets with IID and non-IID distributions.
> |  Method  | Fed-SLAKE | Fed-SLAKE | Fed-FGVC | Fed-FGVC |
> |:-|:-:|:-:|:-:|:-:|
> |  |  IID   | Non-IID |   IID  | Non-IID |
> | Central  |  0.673 |    —    |  0.561 |    —    |
> | FedAvg   |  0.621 |   0.615 |  0.504 |  0.401  |
> | FFA-LoRA |  0.622 |   0.616 |  0.514 |  0.402  |
>
> Table A7:Performance comparison of Mono-InternVL-3B on Fed-Nature dataset. MT-Central refers to centralized training on the centralized multi-task dataset.
> | Mode | VQA | Caption Generation | Visual Grounding | Classification |
> |:-|:-:|:-:|:-:|:-:|
> | MT-Central| 0.686 | 1.020/0.398 | 0.326  | 0.911 |
> | FedAvg    | 0.711 | 1.080/0.406 | 0.337  | 0.930 |
> | FFA-LoRA  | 0.663 | 1.037/0.398 | 0.312  | 0.906 |
>
> Table A8: Performance comparison of Qwen2.5-VL-7B on two representative single-task datasets with IID and non-IID distributions.
> | Method  | Fed-SLAKE | Fed-SLAKE | Fed-FGVC | Fed-FGVC |
> |:-|:-:|:-:|:-:|:-:|
> | |  IID  |  Non-IID  |   IID  | Non-IID |
> | Central  |  0.808  |      — |   0.913  |      — |
> | FedAvg   |  0.781  |  0.785   |   0.885 |  0.851  |
> | FedProx  |  0.768  |  0.766   |   0.880 |  0.843  |
> | FFA-LoRA |  0.746  |  0.750   |   0.869 |  0.796  |
>
> Table A9:Performance comparison of Qwen2.5-VL-7B on Fed-Nature dataset. MT-Central refers to centralized training on the centralized multi-task dataset.
> | Mode | VQA | Caption Generation | Visual Grounding | Classification |
> |:-|:-:|:-:|:-:|:-:|
> | MT-Central| 0.823 |  1.545/0.463    |  0.675  |  0.982  |
> | FedAvg    | 0.820 |  1.542/0.462    |  0.647  |  0.981  |
> | FFA-LoRA  | 0.791 |  1.537/0.457    |  0.598  |  0.976  |
>
> [R1]Luo G, et al. Mono-internvl: Pushing the boundaries of monolithic multimodal large language models with endogenous visual pre-training. CVPR. 2025.

---

> ### Author Response · Authors · 2025-11-25
> **Response to Reviewer yqY5**
>
> **Q4. Key factors that the FL community cares about, such as heterogeneous environments and varying numbers of clients, are not well explored or reflected in the benchmark.**
>
> We thank the reviewer for this insightful comment. We fully agree that system heterogeneity (heterogeneous environments and varying numbers of clients) is a key practical challenge in FL. While the primary focus of our work is to establish a foundational benchmark for the novel and critical problem of data heterogeneity in FL+VLM, we have conducted preliminary experiments to illustrate the extensibility of our framework to system heterogeneity.
>
> Specifically, we conduct additional experiments on two representative datasets: Fed-SLAKE (text-driven) and Fed-FGVC (vision-centric). For both datasets, we partition the data into 50 clients in an IID manner, ensuring that each label was evenly distributed across clients. Following prior works [R2-R3], we simulate heterogeneous FL environments by introducing "stragglers," defined as clients unable to complete local training within a set communication deadline. The straggler ratio quantifies the proportion of selected clients that fail to participate in model aggregation per round. For example, a straggler ratio of 0% indicates a fully homogeneous environment where all selected clients successfully complete their training, while a ratio of 10% means that, on average, 10% of selected clients fail to finish training in each round.
>
> As shown in Tab.A10, for the text-driven Fed-SLAKE dataset, we observe that increasing the straggler ratio from 0.1 to 0.7 only modestly increases the communication cost (e.g., 27/25×). In contrast, for the vision-centric Fed-FGVC dataset, the communication cost rises more sharply (e.g., 38/25×). This trend is consistent with our previous observations regarding data heterogeneity: text-driven tasks tend to benefit from the inherent linguistic priors of LLMs, making them more robust to system heterogeneity; whereas vision-centric tasks, which rely more heavily on fine-tuning the connector and LLMs, show greater vulnerability to uneven client participation.
>
> These experiments demonstrate that FedVLMBench is readily extensible to more realistic FL scenarios. A comprehensive study of the combined effects of system and data heterogeneity is a promising direction for future work. We have included these results and the discussion in the Appendix.
>
> Table A10: Total transmitted message size (number of communication rounds × model parameters in millions) required by FedAvg to reach the target performance (accuracy 0.7 for Fed-SLAKE and 0.55 for Fed-FGVC) under varying straggler ratios, simulating heterogeneous environments. Results are reported for encoder-based VLMs with LLAMA3.2-3B as the baseline.
>
> | Struggler\% | Fed-SLAKE(0.7) | Fed-FGVC(0.55) |
> |:-|:-|:-:|
> |0   | 24 x 23.9 |  25 x 23.9  |
> |10% | 25 x 23.9 |  25 x 23.9  |
> |30% | 26 x 23.9 |  27 x 23.9  |
> |50% | 28 x 23.9 |  34 x 23.9  |
> |70% | 27 x 23.9 |  38 x 23.9  |
>
> [R2]Lai F, et al. Fedscale: Benchmarking model and system performance of federated learning at scale. PMLR, 2022.
>
> [R3]Chen D, et al. Fs-real: Towards real-world cross-device federated learning. KDD. 2023.

---

### Official Review · Reviewer_J2Ci · 2025-10-30

**Soundness:** 3
**Presentation:** 3
**Contribution:** 3
**Rating:** 6
**Confidence:** 5

**Summary:**

This paper introduces FedVLMBench, a comprehensive benchmark for federated fine-tuning of Vision-Language Models. The benchmark systematically evaluates two VLM architectures (encoder-based and encoder-free), four fine-tuning strategies, six federated learning algorithms, and six multimodal datasets spanning single-task and multi-task scenarios across four downstream task categories. Through extensive experiments, the paper investigates the interplay between architectural choices, fine-tuning strategies, and data heterogeneity, providing empirical guidelines for federated VLM deployment in privacy-sensitive domains.

**Strengths:**

- The experiments are comprehensive, covering comparisons across model architectures, fine-tuning methods, and heterogeneity levels.

- The paper is well-organized and clearly written, with strong visual support through well-designed figures and tables.

**Weaknesses:**

- While incorporating both encoder-based and encoder-free architectures enhances the comprehensiveness of the benchmark, this inclusion appears to be driven by empirical considerations rather than theoretical motivation. The introduction briefly states that existing FL studies mainly focus on encoder-based VLMs, and that encoder-free architectures have recently emerged, but it does not sufficiently justify why this comparison is fundamentally necessary in the federated context. The authors do not clearly provide a conceptual analysis of how the two architectures fundamentally differ under FL constraints.

- In Table 3, the linear layer does not consistently outperform the 6-layer MLP, which slightly contradicts the statement in Section 5.2. It is recommended that the authors moderate this claim.
- In Section 5.2, the comparison between F-CL and F-2stage lacks detailed analysis. As shown in Figure 4, the two strategies also exhibit distinct behaviors in text-dominant and vision-dominant tasks. It would be beneficial to further analyze the results from the perspective of modality dominance.

**Questions:**

Please refer to the weaknesses part.

---

> ### Author Response · Authors · 2025-11-25
> **Response to Reviewer J2Ci**
>
> We sincerely thank the reviewer for the efforts and positive feedback. We are pleased that the experiments behind FedVLMBench are viewed as "comprehensive" and "with strong visual support through well-designed figures and tables". Please see our response below regarding the specific comments.
>
> **Q1. It does not sufficiently justify why the comparison of encoder-based and encoder-free architectures is fundamentally necessary in the FL context and how the two architectures fundamentally differ under FL constraints in FL.**
>
> We sincerely thank the reviewer for the insightful comment, which has helped us clarify the core objective of our work. The reviewer is correct that our inclusion of both encoder-based and encoder-free architectures is driven by practical considerations. However, we wish to emphasize that this is not a shortcoming but a fundamental design choice aligned with the primary goal of our benchmark.
>
> Our core study is **not a head-to-head comparison** to see which VLM architecture is "better." Instead, we hope to establish the first systematic benchmark for federated fine-tuning of VLMs. A high-quality and practical benchmark must reflect the reality of the field.  Currently, the community predominantly employs two distinct VLM architectural paradigms: encoder-based and encoder-free VLMs. These architectures differ fundamentally in their components (e.g., the former involves a vision encoder, a connector, and an LLM, while the latter does not) and thus necessitate different fine-tuning strategies. FL researchers and practitioners need guidance on how to effectively apply FL, **regardless of the VLM architecture they adopt**. By studying both, we ensure that **FedVLMBench serves as a universally relevant resource, rather than one limited to a specific architectural niche**.
>
> We deeply appreciate this comment. To avoid misunderstanding, we have revised the introduction in our manuscript to clearly state our core objective and to better justify the inclusion of both architectures based on their prevalence and practical significance (see L52-L82 in the revised manuscript).
>
> **Q2. The linear layer does not consistently outperform the 6-layer MLP in Table 3, which slightly contradicts the statement in Section 5.2.**
>
> We thank the reviewer for this careful observation. We agree that our initial wording was overly absolute. In response, we have now revised the statement in Section 5.2 to accurately reflect the experimental data. The original sentence in lines 325-327 is modified to:
> "In the majority of experimental settings (10 out of 16 cases), both the simple linear layer and the 2-layer MLP achieve comparable or superior performance to the more complex 6-layer MLP." Nevertheless, this revision maintains our core finding that increased connector complexity does not reliably yield performance gains. Furthermore, given the higher communication cost associated with transmitting the 6-layer MLP in each FL round, the 2-layer MLP remains a more practical and efficient choice. Therefore, this adjustment does not affect our proposed Takeaway 1.

---

> ### Author Response · Authors · 2025-11-25
> **Response to Reviewer J2Ci**
>
> **Q3. As shown in Table 4, the F-CL and F-2stage strategies also exhibit distinct behaviors in text-dominant and vision-dominant tasks. It would be beneficial to further analyze the results from the perspective of modality dominance.**
>
> We thank the reviewer for their insightful observation and valuable suggestion. To further explore this phenomenon, we conduct additional experiments on another vision-centric dataset, Fed-RadGenome. As shown in Tab.A1, F-2stage generally achieves slightly better performance than F-CL on vision-centric tasks (Fed-RadGenome and Fed-FGVC), whereas F-CL demonstrates a marginal advantage on text-centric tasks (Fed-SLAKE and Fed-ScienceCap). This pattern can be explained as follows: for vision-centric tasks, F-2stage prioritizes connector optimization, which allows for more precise adaptation of visual feature extraction. In contrast, for text-centric tasks, F-CL leverages the dynamic co-adaptation between the connector and the LLM, leading to better integration of textual information.
>
> Nevertheless, considering the practical constraints of FL, such as communication costs (as shown in Tab.A2), F-CL offers a more balanced and communication-efficient strategy, making it preferable for general deployment .
>
>
> We greatly appreciate the reviewer’s suggestion, which prompted this deeper analysis. We have incorporated these additional discussions into the revised manuscript.
>
> Table A1: Quantitative comparison of four fine-tuning strategies on multi-type task dataset with IID and non-IID distributions.
> |  Mode  |  Method  | Fed-SLAKE | Fed-SLAKE |Fed-ScienceCap | Fed-ScienceCap |Fed-FGVC | Fed-FGVC |Fed-RadGnome | Fed-RadGnome |
> |:-|:-:|:-:|:-:|:-:|:-:|:-:|:-:|:-:|:-:|
> |      |  |   IID  | Non-IID | IID  | Non-IID | IID  | Non-IID | IID  | Non-IID |
> | F-CL     | Central|  0.839 |    -    |  7.550/0.901 |      -      |0.764 |   -   |0.635 |   -   |
> | F-CL     | FedAvg |  0.823 |  0.827  |  7.501/0.898 | 7.476/0.897 |0.721 | 0.603 |0.616 | 0.535 |
> | F-2stage | Central|  0.845 |    -    |  7.591/0.908 |    -        |0.772 |    -  |0.648 |    -  |
> | F-2stage | FedAvg |  0.811 |  0.814  |  7.334/0.884 | 7.281/0.883 |0.730 | 0.614 |0.621 | 0.544 |
>
> Table A2: Total transmitted message size (number of communication rounds and model parameters transmission (GB) in all) required by FedAvg to reach the target performance (accuracy 0.8 for Fed-SLAKE and 0.7 for Fed-FGVC) based on encoder-based VLM (LLaVA-1.5-3B).
> |Mode|Method| Fed-SLAKE(0.8)  | Fed-FGVC(0.7) |
> |:-|:-:|:-:|:-:|
> | F-CL     | FedAvg |  15 (6.69)  | 33 (14.72) |
> | F-2stage | FedAvg |  41 (9.31)  | 72 (16.11) |

---

### Author Response · Authors · 2025-11-26
**General Response to All Reviewer**

We sincerely thank all the reviewers for their insightful comments and constructive suggestions. We greatly appreciate the reviewers' recognition of our efforts, particularly regarding the "comprehensive comparison" and the "pratical guidance" provided for the federated learning community, both of which are incredibly encouraging for us. We are also grateful for the insightful questions and suggestions, which have helped clarify and improve the manuscript. Below, we summarize the main discussion points and the corresponding updates we have made in the revision:


**1. In-depth Analysis of Key Findings**

Following the constructive feedback from the reviewers (yqY5, egwi and Jmzy), we have provided more in-depth investigation into: 1) why text-centric tasks are less sensitive to data heterogeneity than vision-centric tasks? 2) deep exploration of why complex connectors (6-layer MLP) underperform simple ones in encoder-based VLM.

**2. Clarified Motivations & Contributions**

We have clearly summarized our research motivation, the specific gaps in Federated Learning for VLMs, and our three key contributions: 1) A novel and extensive federated VLM dataset suite; 2) A unified public codebase for FL-VLM research; 3) Comprehensive empirical analyses with new findings.


**3. More Expanded Experiments**

We have expanded our experiments to include: 1) Enhanced generalizability validation of our key findings by including two additional open-source VLMs (Reviewer yqY5 and Jmzy) and Non-IID experimental results for Tab.3 (Reviewer egwi); 2) Evaluation under realistic system heterogeneity settings (Reviewer yqY5); 3) Adding support with one more recent FL algorithm (FedSA-LoRA, ICLR 2025) (Review egwi); 4) Communication cost analysis between joint finetuning and sequential fine-tuning strategies (Reviewer J2Ci).

**4. More Researcher-friendly Illustrations**

Considering the recommendations highlighted by the reviewer egwi, we have provided a clear visual figure of various FL methods in Appendix to enhance accessibility for researchers from both VLM and medical domains.

We hope this summary clarifies the revisions we have made. These modifications, along with the updated tables and insights, are included in our revised manuscript (PDF file, with modifications marked in blue). We believe that the manuscript has been significantly strengthened thanks to your insightful comments. Thanks again to the reviewers for your time and effort in reviewing our paper.

---

### Meta-Review · Area_Chair_tsSJ · 2026-01-07

**Summary:**

Reviewers generally view the benchmark as well organized, and empirically comprehensive, noting its broad coverage of VLM architectures, downstream tasks, fine-tuning strategies, and federated learning settings. The inclusion of medical-domain scenarios and multi-task FL is considered practically meaningful, and the experimental results provide useful empirical guidelines for federated fine-tuning of VLMs. However, several common concerns are raised. The work is seen as largely descriptive and empirically driven, with limited technical or theoretical innovation and insufficient conceptual motivation—particularly regarding why encoder-based and encoder-free architectures should behave differently under FL constraints. Performance differences among fine-tuning strategies are often small, weakening some of the stated conclusions, and certain claims are not fully supported by the reported results. Reviewers also question the breadth and representativeness of the benchmark, citing limited model diversity, small client counts, and insufficient exploration of key FL factors such as non-IID heterogeneity, fairness, and scalability. Overall, while FedVLMBench is regarded as a useful resource-style contribution, reviewers suggest that stronger motivation, deeper analysis, broader model and FL settings, and clearer justification of the main takeaways are needed to elevate it beyond a technical benchmark report. Therefore, AC's recommendation is to reject.

**Reviewer Concerns:**

During the rebuttal phase, the author presented extensive experimental results to address concerns regarding the breadth and representativeness of the benchmarks. The author also clarified that the reason for introducing the encoder-free architecture was to provide researchers with a more systematic experimental benchmark. However, there has been no substantial progress in addressing issues where technological or theoretical innovation is limited. The authors claim to present “newly discovered comprehensive and in-depth empirical analysis,” yet they fail to provide guiding analytical conclusions for designing VLM federated learning algorithms. The author describes this as a “comprehensive benchmark tailored for the emerging field of FL-VLM,” yet the exploration of key FL factors has been relegated to future work, which was heavily questioned by reviewers.

**Reviewer Scores:**

I expect the final rating to be as follows:
- Reviewer J2Ci: 6
- Reviewer yqY5: 4
- Reviewer egwi: 6
- Reviewer Jmzy: 2

---

### Decision · Program_Chairs · 2026-01-26

Reject